# Spatial patterns of microbial communities across surface waters of the Great Barrier Reef

Pedro R. Frade [1,2 ✉], Bettina Glasl[2,3], Samuel A. Matthews[3,4], Camille Mellin [5,6], Ester A. Serrão [1], Kennedy Wolfe[7,8], Peter J. Mumby[7,8], Nicole S. Webster [3,9] & David G. Bourne[2,3]

Microorganisms are fundamental drivers of biogeochemical cycling, though their contribution to coral reef ecosystem functioning is poorly understood. Here, we infer predictors of bacterioplankton community dynamics across surface-waters of the Great Barrier Reef (GBR) through a meta-analysis, combining microbial with environmental data from the eReefs platform. Nutrient dynamics and temperature explained 41.4% of inter-seasonal and cross-shelf variation in bacterial assemblages. Bacterial families OCS155, Cryomorphaceae, Flavobacteriaceae, Synechococcaceae and Rhodobacteraceae dominated inshore reefs and their relative abundances positively correlated with nutrient loads. In contrast, Prochlorococcaceae negatively correlated with nutrients and became increasingly dominant towards outershelf reefs. Cyanobacteria in Prochlorococcaceae and Synechococcaceae families occupy complementary cross-shelf biogeochemical niches; their abundance ratios representing a potential indicator of GBR nutrient levels. One Flavobacteriaceae-affiliated taxa was putatively identified as diagnostic for ecosystem degradation. Establishing microbial observatories along GBR environmental gradients will facilitate robust assessments of microbial contributions to reef health and inform tipping-points in reef condition.

[1] Centre of Marine Sciences, University of Algarve, Faro, Portugal. [2] College of Science and Engineering, James Cook University, Townsville, QLD, Australia. [3] Australian Institute of Marine Science, Townsville, QLD, Australia. [4] ARC Centre of Excellence in Coral Reef Studies, James Cook University, Townsville, QLD, Australia. [5] Institute for Marine and Antarctic Studies, University of Tasmania, Hobart, TAS, Australia. [6] The Environment Institute and School of Biological Sciences, University of Adelaide, Adelaide, SA, Australia. [7] Marine Spatial Ecology Lab, School of Biological Sciences, University of Queensland, St Lucia, QLD, Australia. [8] ARC Centre of Excellence for Coral Reef Studies,  University of Queensland, St Lucia, QLD, Australia. [9] Australian Centre for Ecogenomics, University of Queensland, Brisbane, QLD, Australia. ✉email: pedro.frade@nhm-wien.ac.at

Coral reef ecosystems are under increasing anthropogenic pressure leading to widespread global degradation[1]. A combination of global disturbances, such as rising seawater temperatures[2], and local pressures including overfishing, declining water quality, disease and outbreaks of coral predating crown-of-thorns starfish[3], are driving declines in coral condition. Despite being considered among the best-managed marine areas, parts of the Great Barrier Reef (GBR) are threatened by nutrient and sediment inputs from land-based sources[4,5]. The GBR was also impacted by back-to-back bleaching events in 2016–2017, resulting in the mortality of one-third of all its shallow-water corals[6] and dramatic impairment of recruitment capacity of the surviving coral[7]. Another mass bleaching was reported in early 2020, representing three events in just 5 years, each with extensive spatial impacts on the ecosystem values of the GBR[8]. There is currently an urgent requirement to better understand the functioning of coral reefs and identify the factors underpinning their resilience or susceptibility, to determine how the socio-economic and ecological value of these ecosystems will change.

Microorganisms are fundamental drivers of biogeochemical cycling in coral reef waters[9,10] and a crucial component of the coral holobiont[11]. However, their contribution to the functioning and resilience of reefs is not well understood[12–14]. Shifts in the compositional and functional diversity of both coral-associated[15,16] and free-living planktonic[12,17] microbial communities have been linked to varying levels of anthropogenic impact, including changes in seawater nutrient levels[18]. For example, chronic nutrient exposure has been correlated with increased prevalence of coral disease in Caribbean reef systems[19]. Protection from fishing has led to improved reef health through the promotion of microbial diversity as opposed to the growth and rapid development of opportunistic microbial pathogens in reefs open to fishing pressures[20]. These findings were validated in a field experiment simulating overfishing and nutrient pollution, which interacted with sea surface temperatures to drive changes in coral microbiomes and an increase in coral mortality[21]. The health and condition of corals, and resilience of reefs to environmental stressors more broadly, is inherently linked to microbial functioning in these ecosystems.

Pressures such as overfishing and nutrient pollution can ultimately contribute to the top down and bottom up processes that drive phase-shifts from coral-dominated to macroalgal-dominated reef ecosystems[22,23]. These phase-shifts are reinforced by the positive feedback loop proposed in the DDAM model (DOC, disease, algae, microorganism), through which macroalgae-derived labile dissolved organic carbon supports copiotrophic and potentially pathogenic bacterioplankton communities that harm corals, therefore promoting algal competitive dominance[24,25]. The concomitant increase in microbial abundances on algal-dominated reefs worldwide results in a switch from autotrophic to heterotrophic microbial processes and a shift in trophic structure towards higher microbial biomass and energy use, a phenomenon coined microbialization[26,27]. An additional hypothesis considers the self-reinforcement of macroalgal dominance through microbial pathways. Specifically, positive microbial responses to the photosynthates leached from algae may increase the vertical attenuation of light, thereby suppressing coral calcification and elevating stress on the coral[28].

Shifts in free-living microbial lineages in response to seawater nutrient gradients and benthic composition in reef systems have previously been reported. For example, atolls of the Line Island chain that experience the highest levels of coral disease, nitrogen and phosphate, and lowest coral cover also display microbial abundances that are tenfold higher and communities dominated by heterotrophs, including a large percentage of potential pathogens[12], features characteristic of near-shore

environments. In addition, the types of bacterial autotrophs changed from *Prochlorococcus*-dominated assemblages in the most pristine regions to *Synechococcus*-dominated communities at atolls with human-influences such as increased concentrations of nitrogen and phosphate[12]. Higher nutrient availability also enriched for nutrient-related microbial metabolic traits such as nitrate and nitrite ammonification[29]. In the Caribbean, microbial signatures were also clearly distinct between protected and offshore Cuban reefs compared to human impacted reefs in the Florida Keys[18]. Similarly, a study across three ocean basins observed that algal-dominated sites were enriched in copiotrophic microbial taxa, including Gammaproteobacterial families such as Enterobacteriaceae, Vibrionaceae, Shewanellaceae and Pasteurellaceae, and Bacteroidetes such as Cytophagaceae and Flavobacteriaceae, whereas coral-dominated reefs were enriched in oligotrophic Alphaproteobacterial families such as Caulobacteriaceae, Sphingomonadaceae, Hyphomonadaceae, Bradyrhizobiaceae, Acetobacteriaceae, Phyllobacteriaceae, Rhodospirillaceae, Pelagibacteraceae, Rhizobiaceae and Rhodobacteriaceae[26].

Temporal and environmental variability in microbial assemblages at specific GBR sites have been identified, however the drivers of bacterioplankton community change are poorly resolved across the large expanse of the GBR. In the northern GBR (Tully River region), microbial communities in proximity to reefs followed seasonal dynamics and responded to riverine inputs, with rainfall, water quality (i.e., nutrient, organic compounds and herbicides), salinity and temperature implicated as drivers of bacterial community composition[30]. In the southern GBR, along cross-shelf gradients in the Mackay region, bacterioplankton numbers correlated with dissolved organic carbon and particulate carbon, nitrogen and phosphorus[31]. Seawater microbiomes from inshore reefs in the central GBR were investigated for their predictive power to identify environmental perturbations affecting reefs, with bacterial compositional variability significantly explained by temperature in addition to water quality parameters such as total suspended solids, particulate organic carbon or chlorophyll concentrations[32]. Current evidence from a number of global studies indicates that cumulative environmental pressures and the ability of microbial communities to tolerate and respond to various abiotic and biotic conditions structure the microbial community[33]. However, the lack of available microbial data collected at sufficient spatial and temporal resolution, and supported by a comprehensive suite of contextual parameters, limits our understanding of the role of microbes in the functioning and resilience of the GBR and coral reef ecosystems more generally[11].

In recent years, the need to scale-up management of natural ecosystems has been increasingly aided by modelling efforts. For instance, ecosystem models have inferred relationships between long-term anthropogenic pollution and reef resilience[34] and mapped their impacts on coral reef communities across the continental shelf[35]. Models of water quality such as those integrated in the eReefs collaborative data platform are now available (https://ereefs.org.au/), allowing for interpretation and prediction of ecosystem patterns at large temporal and spatial scales. As contributors to coral reef functioning, it is important that microbial communities are incorporated into modelling frameworks to elucidate their applicability for ecosystem management and bioremediation[36,37]. A first step towards a global understanding of microbial communities in the context of ecosystem functioning in the GBR is to evaluate how much we know and, importantly, identify knowledge gaps in microbial dynamics across this threatened ecosystem[38].

Here, we present an original meta-analysis targeting compositional variation of microbial communities in GBR waters and

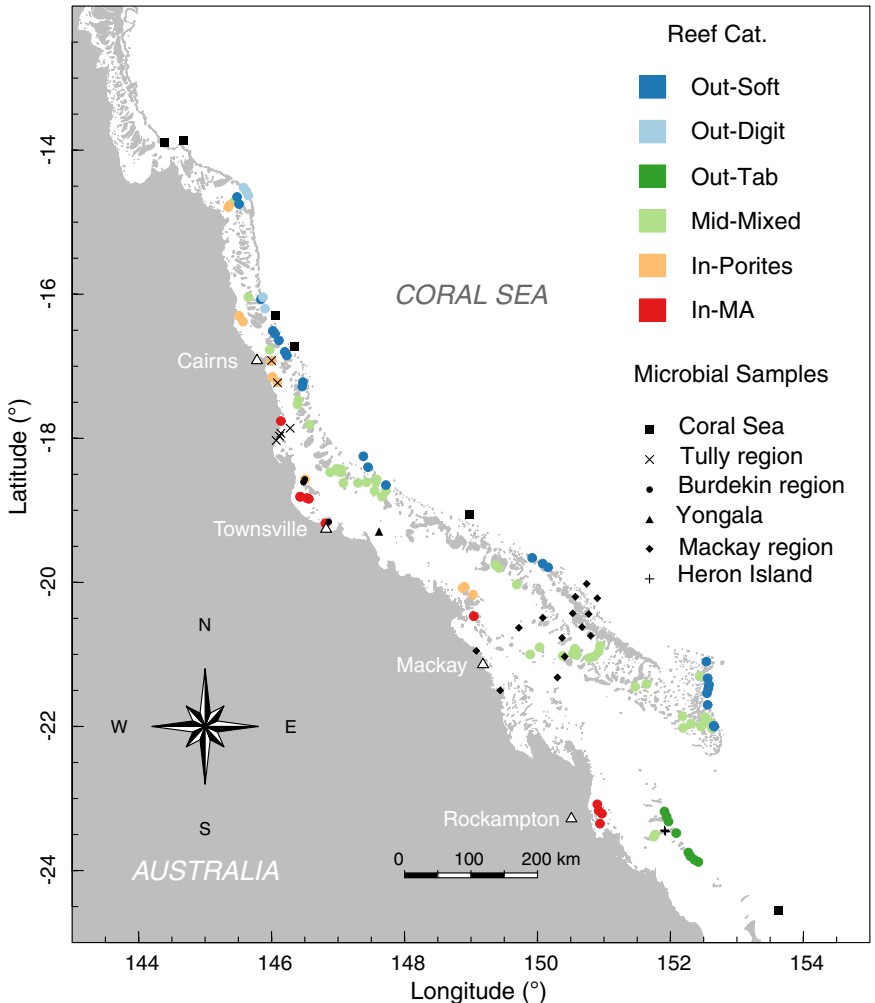

**Fig. 1 Distribution of microbial samples and environmental datasets used in this study.** Map of the Great Barrier Reef showing sites with available microbial data (case studies, in black) and reefs included in the Long-Term Monitoring Program (LTMP, colour coded) for which comprehensive abiotic data are available from the eReefs platform. Colour-coded reef categories (sensu Mellin et al.[35]) are: (1) Out-Soft—outershelf soft coral communities (dark blue), (2) Out-Digit—outershelf branching hard coral (light blue), (3) Out-Tab: outershelf tabular and corymbose hard coral (dark green), (4) Mid-Mixed—midshelf turf algae communities (light green), (5) In-Porites— inshore hard coral communities (orange), and (6) In-MA—inshore macroalgae communities (red). Sites representative of microbial case studies are indicated by squares (Coral Sea), circles (Burdekin region, available in Glasl et al.[32]) and triangles (Yongala location), all of which are BioPlatforms Australia (BPA) datasets, diagonal crosses (Tully region, available in Angly et al.[30] dataset), diamonds (Mackay region, available in Alongi et al.[31] dataset) and plus signs (Heron Island, available in Epstein et al.[40] dataset). Please see Supplementary Table 1 for a summary of microbial datasets used in this study.

identify putative taxonomic (and functional) groups of reef microbes across known environmental gradients and for distinct reef categories. We use the coral community mapping of Mellin et al.[35] as the categorization framework. This classification splits reefs surveyed across the GBR as part of the Long-Term Monitoring Program (LTMP) of the Australian Institute of Marine Science (AIMS)[39] into six main reef benthic categories according to environmental predictors (such as distance to the barrier reef edge, seasonal range in seabed oxygen concentration and temperature, seasonal range in sea surface temperature, and percentage of carbonate sediments) (see Fig. 1). These reef benthic categories differ in their macroorganism indicator taxa and geographic position across the GBR shelf and are represented as: (1) outershelf soft coral communities (Out-Soft), (2), outershelf branching hard coral (Out-Digit), (3) outershelf tabular and corymbose hard coral (Out-Tab), (4) midshelf turf algae communities (Mid-Mixed), (5) inshore hard coral communities (In-Porites) and (6) inshore macroalgae communities (In-MA).

## Results

**Environmental variation across GBR surface waters**. Modelled estimates of the environmental conditions of surface seawater retrieved from the eReefs hydrodynamic and biogeochemical model (GBR1, https://research.csiro.au/ereefs/models/model-outputs/gbr1/) for the microbial case study ($n = 37$) and LTMP ($n = 109$) sites (see Fig. 1), covered 16 environmental variables known as potential drivers of microbial community variation (see Fig. 2 and Supplementary Fig. 1). Reefs within distinct categories differed considerably in their prevalent surface seawater conditions (Fig. 2). Overall, organic and inorganic nutrients decreased in concentration with increasing distance from the shore (Fig. 2 and Supplementary Fig. 1; In-MA and In-Porites > Mid-Mixed and Out-Tab > Out-Soft and Out-Digit). The exceptions were the inorganic nitrogen variables (DIN, NH4 and NO3), which, together with chlorophyll a, peaked at midshelf reefs, particularly in the Out-Tab reef category. Superimposed on this inshore to outershelf trend, there was strong seasonal variation. However,

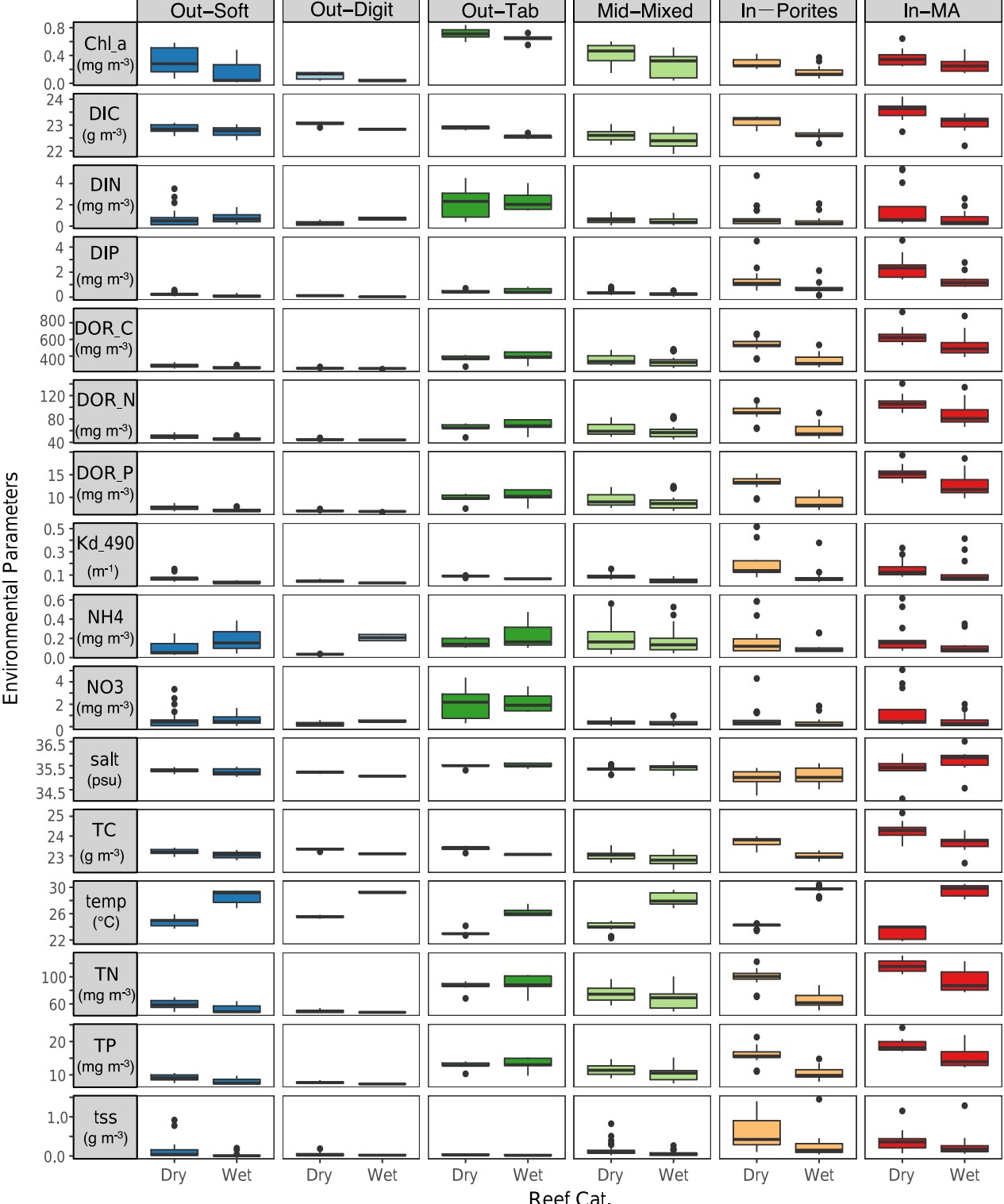

**Fig. 2 Cross-shelf and inter-seasonal environmental variation (boxplots) across the Great Barrier Reef (GBR) for parameters retrieved from the eReefs platform.** Data based on 146 LTMP and microbial sites combined (*n* = 4672 independent modelling experiments). Chl_a: total chlorophyll *a*, DIC: dissolved inorganic carbon, DIN: dissolved inorganic nitrogen, DIP: dissolved inorganic phosphorus, DOR_C: dissolved organic carbon, DOR_N: dissolved organic nitrogen, DOR_P: dissolved organic phosphorus, KD_490: vertical attenuation coefficient of light at 490 nm, NH4: ammonium, NO3: nitrate, salt: salinity, TC: total carbon, temp: temperature, TN: total nitrogen, TP: total phosphorus, and tss: total suspended solids. Colour coded reef categories (sensu Mellin et al.[35]) as detailed in Fig. 1.

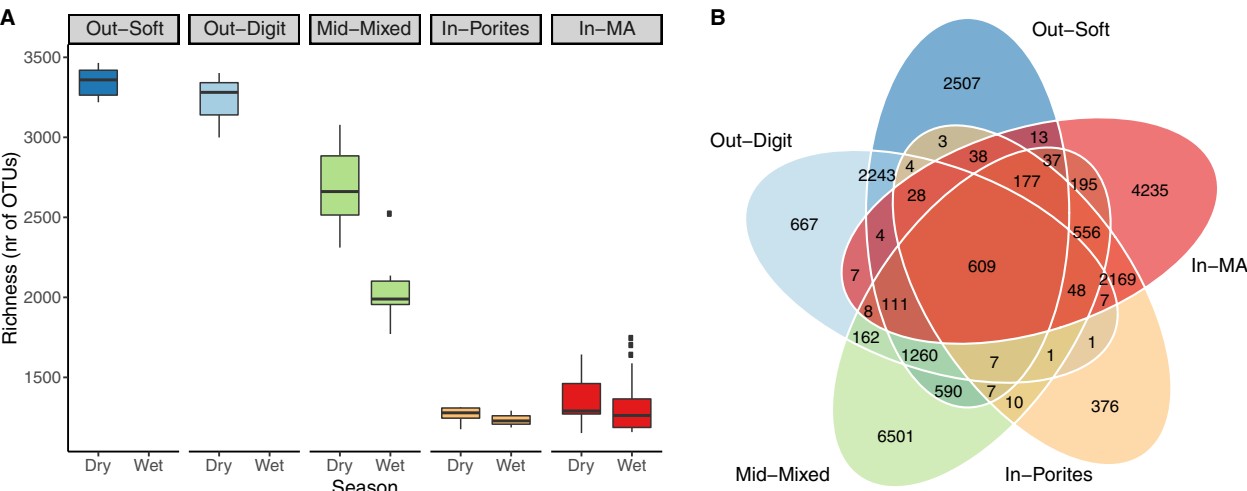

**Fig. 3 Microbial community diversity across reef categories of the Great Barrier Reef (GBR). a** Richness (observed OTU number per sample) across reef category and season (please see Supplementary Fig. 4 for other alpha-diversity indexes). **b** Number of unique, shared and ubiquitous OTUs across reef categories. $n = 69$ biologically independent samples. Colour coded reef categories (sensu Mellin et al.[35]) as detailed in Fig. 1. Data for wet season in Out-Soft and Out-Digit not available.

this effect tended to lose its influence towards outershelf categories (Out-Soft and Out-Digit). Out-Soft and Out-Digit were devoid of seasonal effects, with the exception of temperature differences between the wet and dry seasons. Inshore reef categories in contrast, showed strong seasonal differences for all variables measured.

After dimensional reduction based on pairwise correlations between variables (Supplementary Fig. 2), environmental variation represented in six non-collinear variables was modelled by linear discriminant analysis (LDA) to predict reef category for the sites with available microbial community data (Supplementary Fig. 3). Training the LDA model with the LTMP dataset resulted in a weighted average model accuracy of 73.4% (i.e. correct classification of observed reef category as measured by "leave-one-out" cross validation; see Supplementary Fig. 3a) and a Cohen's weighted Kappa of 68.8%, thus allowing assignment of a putative reef benthic community category to the sites included in the microbial dataset (Supplementary Fig. 3b). Nominally, the inshore sites of Mackay and the Burdekin (Magnetic Island) were representative of the In-MA category; the inshore Burdekin sites of Orpheus Island, and Tully (Marine and Plume sites), represent In-Porites; the midshelf sites of Mackay, the Burdekin (Yongala) and Heron Island represent Mid-Mixed; and the Coral Sea sites were representative of the Out-Soft category. Category Out-Tab was represented by a limited number of the outershelf reef sites of Mackay, and category Out-Digit by a small number of the Coral Sea sites (Supplementary Fig. 3b).

**Microbial community assemblages across the GBR.** Locations in the Burdekin region of the GBR (Magnetic Island: $n = 1$ site, Orpheus Island: $n = 2$ sites, Yongala: $n = 1$ site) and the Coral Sea ($n = 6$ sites), which comprised $n = 69$ samples across two seasons (wet and dry), were used to assess broad microbial community assemblages across environmental categories. These sites span the inshore to offshore cross-shelf gradient and cover five out of six defined benthic reef categories (Supplementary Fig. 3b; the exception being the Out-Tab category), putatively representing ~90% of all GBR reefs[35]. In addition, the Tully dataset allowed interpretation of seasonality and plume influence within the single reef category In-Porites, reported as supplementary results. Importantly, microbial community assemblages from the $n = 10$

Burdekin and Coral Sea sites were characterised using identical sequencing protocols and comparable database taxonomic assignment [i.e. generated from microbiome initiative coordinated through BioPlatforms Australia (BPA)[41]]. Alpha diversity (observed richness; Fig. 3a) varied significantly among reef categories and differences were not homogeneous across seasons (significant interaction; $F_{(2,59)} = 14.43$, $p < 0.001$; see Supplementary Table 2 and Supplementary Fig. 4 for further results). Outershelf reefs (Out-Soft and Out-Digit) consistently showed the highest microbial richness, followed by the midshelf reefs (Mid-Mixed), with these categories displaying approximately threefold and twofold higher richness, respectively, than the inshore reefs (In-MA and In-Porites). Microbial richness in each reef category was consistently higher in the dry season than in the wet season. A substantial number of OTUs were unique to a particular reef category (Fig. 3b), with the midshelf reef (Mid-Mixed) having the highest proportion of unique OTUs (63% of all OTUs found in that category), followed by inshore macroalgae communities (In-MA; 51%) and outershelf soft coral communities (Out-Soft; 33%). Outershelf branching hard coral (Out-Digit, only 13% unique OTUs) and inshore hard coral communities (In-Porites, only 9% unique OTUs) shared most OTUs with the other category in their respective reef shelf region, i.e., Out-Soft (with which Out-Digit shared 43% of all OTUs) and In-MA (with which In-Porites shared 54% of all OTUs). Unconstrained ordination (nMDS, Fig. 4a) shows a clear separation of the microbial community based on reef category and some segregation of samples according to season (within each reef category), though the two inshore reef categories (In-Porites and In-MA) and the two outershelf categories (Out-Soft and Out-Digit) have overlapping microbial communities (Fig. 4a). Reef categories (explaining 55% of variation) and season (explaining 5% of variation) significantly structured the surface seawater microbiome, and the effect of reef category was heterogeneous across season (full model PERMANOVA with interaction, pseudo $F_{(9,59)} = 14.04$, $p < 0.001$; see Supplementary Table 3).

Shallow-water pelagic microbiomes in the GBR were dominated by the bacterial phyla Proteobacteria, Cyanobacteria, Bacteroidetes, Actinobacteria and SAR406 (Fig. 4b). The most dominant bacterial families were the cyanobacterial Prochlorococcaceae and Synechococcaceae, and Pelagibacteraceae (Alphaproteobacteria). Inshore reefs (including In-MA and In-Porites)

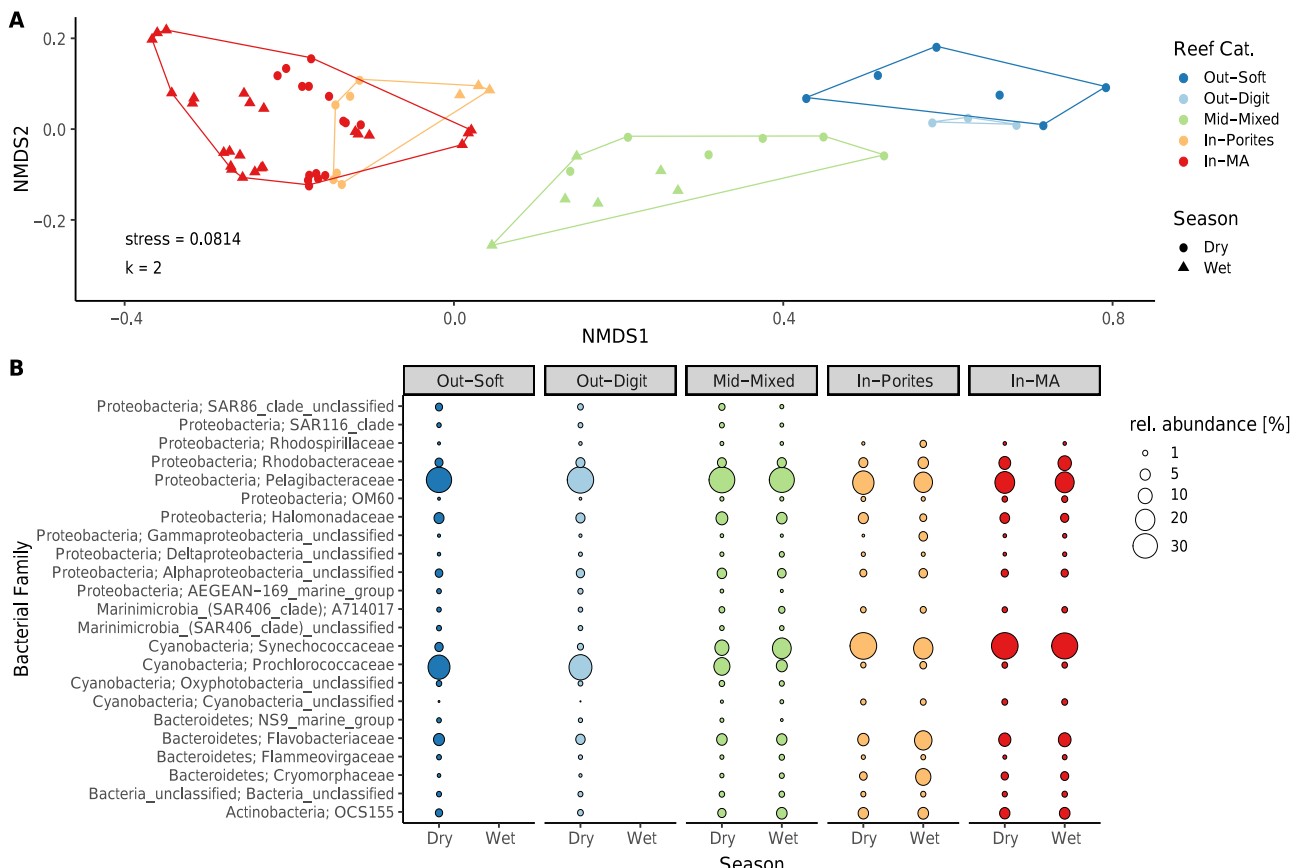

**Fig. 4 Microbial community descriptors across reef categories of the Great Barrier Reef (GBR). a** Unconstrained ordination (nMDS) of GBR microbial communities based on reef category and season. **b** Microbial community composition of dominant bacterial families across GBR reef categories. $n = 69$ biologically independent samples. Colour coded reef categories (sensu Mellin et al.[35]) as detailed in Fig. 1. Data for wet season in Out-Soft and Out-Digit not available.

were dominated by Synechococcaceae. Pelagibacteraceae, Flavo-bacteraceae (Bacteroidetes), Rhodobacteraceae (Proteobacteria), family OCS155 (Actinobacteria), Cryomorphaceae (Bacteroi-detes) and Halomonadaceae (Proteobacteria) (see Fig. 4b) were also abundant. Differences between the two inshore reef categories were minor, but for In-MA, there was a tendency for a higher relative abundance of Synechococcaceae, Rhodobacter-aceae and the typically low-abundance proteobacterial family OM60. Comparatively for In-Porites, there was an increase in Pelagibacteraceae and Prochlorococcaceae. Pelagic microbiomes of Mid-Mixed communities were instead dominated by the Pelagibacteraceae, with Prochlorococcaceae becoming increas-ingly abundant in comparison to the Synechococcaceae, which was the second most dominant taxon (Fig. 4b). These were followed in dominance by the Halomonadaceae, which was enriched in Mid-Mixed communities compared to inshore In-MA and In-Porites reef categories. Mid-Mixed communities also included taxa within the families Flavobacteraceae, Rhodobacter-aceae, OCS155 and Cryomorphaceae which were still abundant, though in contrast to the Halomonadaceae, present at lower relative abundances than for inshore reef categories (In-MA and In-Porites). Pelagibacteraceae also dominated outershelf reef communities (Out-Soft and Out-Digit), and Prochlorococcaceae was the second most abundant family. Halomonadaceae was also highly abundant and enriched in Out-Soft and Out-Digit compared to lower abundances in midshelf and inshore reef categories. Synechococcaceae, Rhodobacteraceae, SAR86 (Proteo-bacteria) and the Flavobacteraceae were next highest in abundance. The family OCS155 showed lower relative abundance

in Out-Soft and Out-Digit compared to all categories further inshore (In-MA, In-Porites and Mid-Mixed), and the Cryomor-phaceae were nearly absent (Fig. 4b). Microbial communities inhabiting Out-Tab pelagic habitats could not be predicted from the available BPA case study data.

**Environmental drivers of microbial community change.** The environmental parameters that influenced microbial communities at sites across GBR surface waters were identified by integrating 16 local environmental variables (derived from eReefs) with community patterns. After dimensional reduction based on pairwise correlations between variables (Supplementary Fig. 5), eight non-collinear variables were visualized by PCA (Supple-mentary Fig. 6a; first two components explained 61.3% of var-iation in the dataset). Variables explaining significant variation in the microbial community were determined by dbRDA (Supple-mentary Table 3). The full model (no interactions considered) was significant according to an ANOVA-like permutational test (pseudo $F_{(7,61)} = 6.15$, $p < 0.01$) and included the variables sali-nity, temperature, total chlorophyll $a$, dissolved inorganic carbon (DIC), ammonium (NH4), dissolved organic carbon (DOC) and the vertical attenuation coefficient of light (Kd_490) (Supple-mentary Fig. 6b). Significant constraints had a total explanatory value of 41.4% of variation, according to Variation Partitioning Analysis. Percentage of variation explained by individual con-straints was 18.3% for DOC, 11.2% for temperature, 4.2% for salinity, 7.6% for NH4, 8.2% for DIC, 3.7% for chlorophyll $a$, and 7.3% for Kd_490. Nutrient dynamics and temperature explained

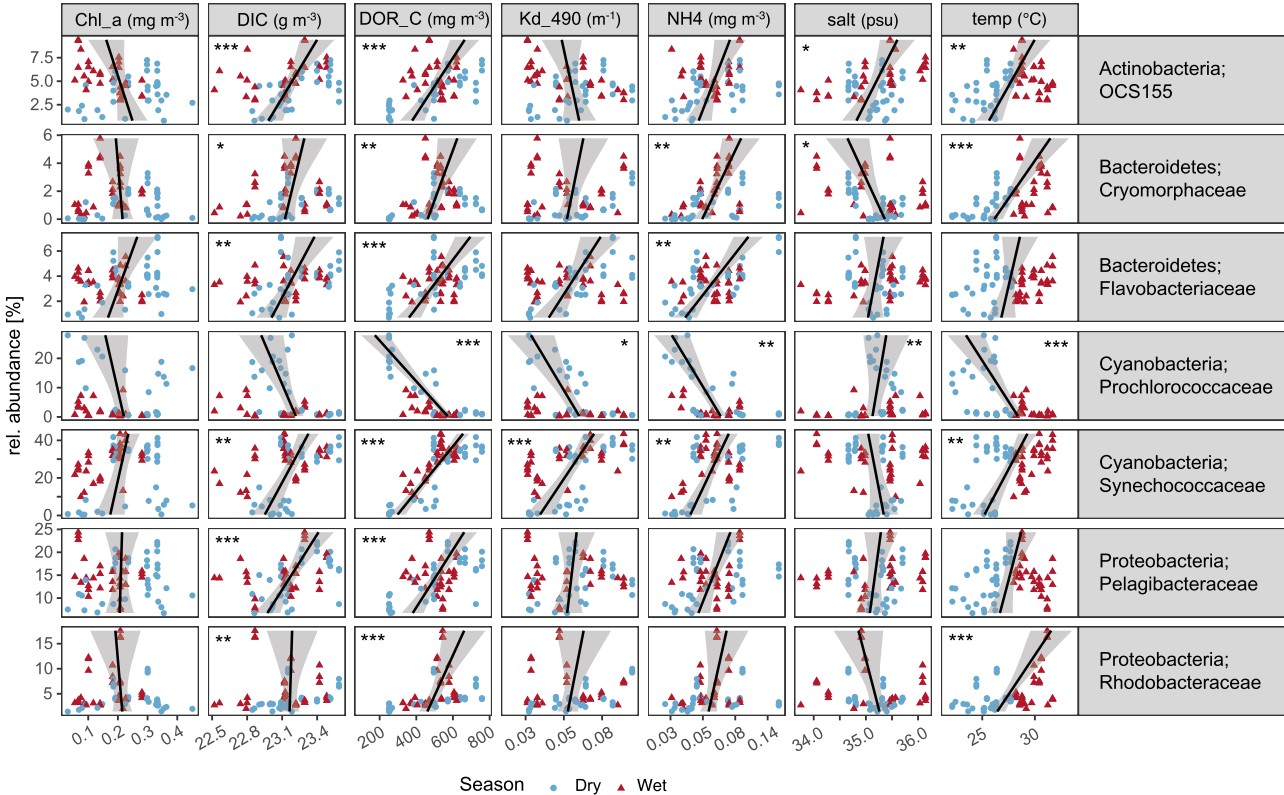

**Fig. 5 Association between individual environmental parameters and the relative abundance of dominant individual bacterial families for the Great Barrier Reef (GBR).** Lines represent fitted linear regression models (*n* = 69 biologically independent samples) and asterisks on upper corners of panels represent adjusted p-values for significance of Spearman correlation coefficients between each environmental parameter and relative abundance of each family (**p* < 0.05, ***p* < 0.01, ****p* < 0.001). Chl_a: total chlorophyll *a*, DIC: dissolved inorganic carbon, DOR_C: dissolved organic carbon, KD_490: vertical attenuation coefficient of light at 490 nm, NH4: ammonium, salt: salinity, and temp: temperature.

most of the variation in the microbial community assemblages. This variation primarily corresponded to differences among reef categories (Supplementary Fig. 6b), influenced by the patterns of organic and inorganic nutrient loads (plus temperature) decreasing from inshore (In-MA and In-Porites), to midshelf (Mid-Mixed) and outershelf (Out-Soft and Out-Digit) reefs. This nutrient and temperature axis also influenced the sample patterns according to season, at least for categories In-MA and Mid-Mixed. The effect of chlorophyll concentration on pelagic microbiome variation related to an increase in attenuation of light with depth, and both these parameters influenced microbial communities in an opposite way to salinity. These variables mostly influenced microbiomes within the two inshore reef categories (In-MA and In-Porites) and the midshelf Mid-Mixed category, but not for outershelf reefs, which displayed more stable community patterns. For the In-Porites and Mid-Mixed reefs, chlorophyll *a*, Kd_490 and salinity explained some of the seasonal variation in microbiome composition (Supplementary Fig. 6b). In contrast, for In-MA reefs, microbiome variation correlated with chlorophyll *a* and salinity changes, though this variation was not structured according to season.

Correlations between individual environmental parameters and individual bacterial families were observed for all GBR reef categories combined (Fig. 5). Increasing nutrient loads (DIC, DOC and NH4) positively correlated with an increase in families OCS155, Cryomorphaceae, Flavobacteriaceae, Synechococcaceae, Pelagibacteraceae and Rhodobacteraceae, with particularly significant correlations during the dry season (Spearman correlation coefficient module often equal or higher than 0.6; Supplementary Fig. 7). In contrast, increasing nutrient loads correlated with

decreasing relative abundances of Prochlorococcaceae. The influence of increased temperature was similar to that of increasing nutrient concentrations across the different bacterial taxa. The exception to this pattern is the negative correlation of temperature with Pelagibacteraceae in the wet season (Supplementary Fig. 7). Synechococcaceae also displayed a response to light attenuation (Kd_490). Salinity correlated significantly (Spearman correlation coefficient, *p* < 0.05) and positively with OCS155 and Prochlorococcaceae, and negatively with Cryomorphaceae (Fig. 5). These environmental-microbial correlations demonstrate that most responses are consistent across seasons (dry versus wet season, but see Supplementary Fig. 7) and capture patterns across broad spatial scales (reef categories and cross-shelf habitats).

**Diagnostic microbes of GBR reef categories**. The inshore reef categories, In-MA and In-Porites, were characterised by a high diversity of microbial indicators (Fig. 6), with 59 and 68 OTUs identified, respectively. The families Rhodobacteraceae and Synechococcaceae were the most dominant indicators though taxa spanning Cryomorphaceae, Flavobacteriaceae, Pelagibacteraceae and Halomonadaceae were also characteristic for these inshore reef categories (see Supplementary Fig. 8 for individual indicator OTUs). The Mid-Mixed reef category was also characterised by a high number of indicators (59 OTUs) though dominated by OTUs affiliated with the Prochlorococcaceae. Outershelf reef categories had far fewer indicator taxa (only 15 OTUs for both Out-Soft and Out-Digit) and were similarly dominated by the Prochlorococcaceae. Members of the OCS155,

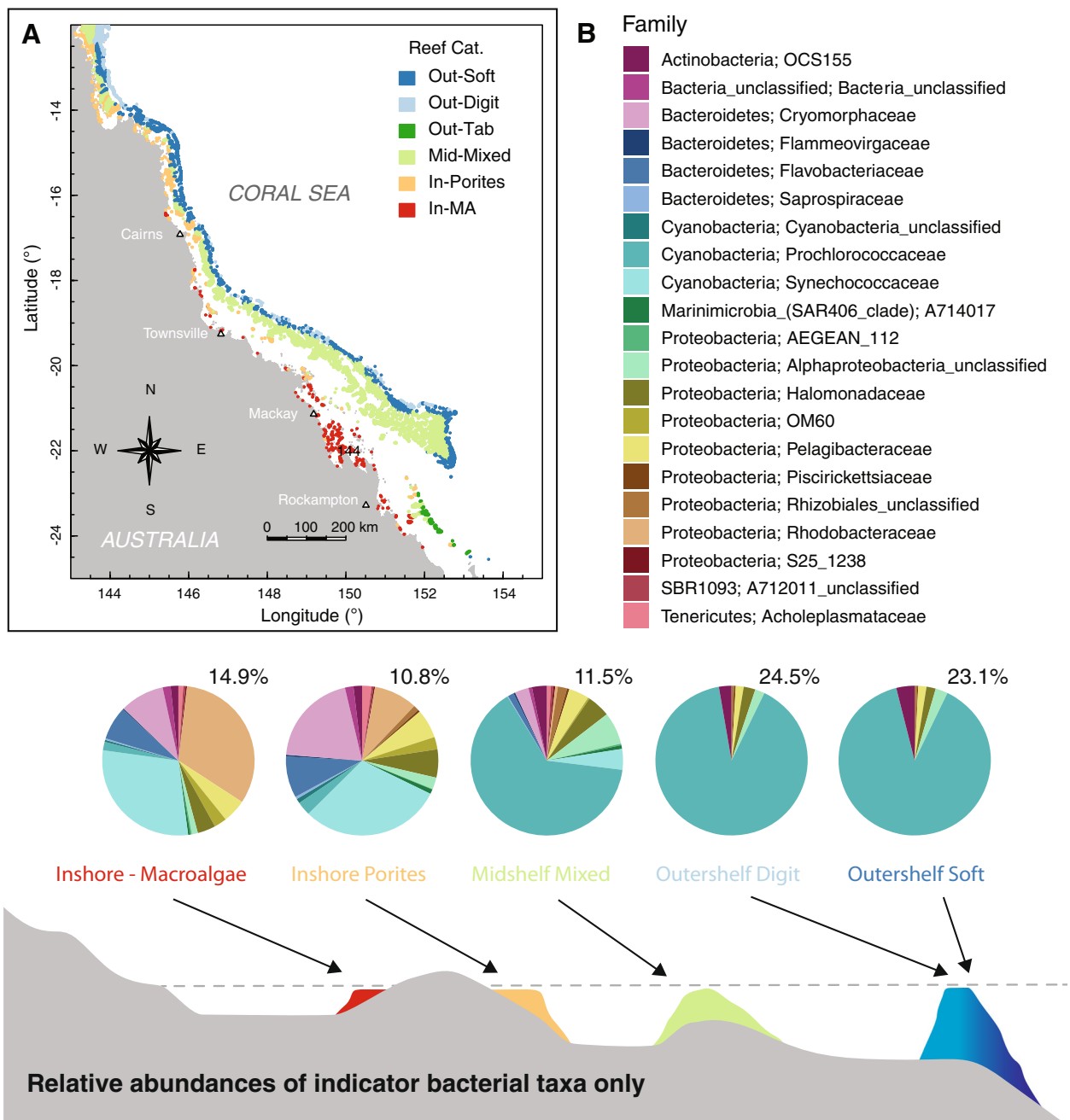

**Fig. 6 Diagnostic microbial communities across surface waters of the Great Barrier Reef (GBR). a** Map of the GBR (adapted from Mellin et al.[35]) showing the spatial distribution of the different reef categories. **b** Microbial indicator taxa of benthic reef categories across the GBR as determined by IndVal analysis (relative abundances at the family level are depicted as a proportion of all indicators recognized for a particular category). Percentages indicate the summed relative abundance of indicator taxa at each reef category, i.e., the total abundance each pie chart represents. Colour coded reef categories (sensu Mellin et al.[35]) as detailed in Fig. 1. Note that panel **a** also includes the microbial sampling sites, now colour-coded according to their modelled reef categories following methodology described in this study; and that the spatial characterisation of the reef in **b** is hypothetical and does not reflect a rigid relative position of the different reef categories.

Halomonadaceae, Pelagibacteraceae and unclassified Alphaproteobacteria comprise the remainder of the indicator taxa in outershelf reefs. It should be noted that when the same family was depicted as an indicator across reef categories, for example OCS155, Pelagibacteraceae and Halomonadaceae (Fig. 6), the specific indicator OTU was different within each reef category (Supplementary Fig. 8).

OTUs identified as an indicator for one particular reef category, were also generally indicators for two or more other reef categories (see Supplementary Fig. 8). For example, 24

different OTUs were identified as indicators of inshore categories (mostly Cryomorphaceae, Flavobacteriaceae, Synechococcaceae, OM60, Pelagibacteraceae and Rhodobacteraceae), 34 OTUs for the combined midshelf and inshore categories (including OCS155, Cryomorphaceae, Flammeovirgaceae, Flavobacteriaceae, Saprospiraceae, Synechococcaceae, Halomonadaceae, OM60, Pelagibacteraceae, Rhodobacteraceae, two families in the SAR406 and Acholeplasmataceae), and 5 OTUs for the combined outershelf and midshelf categories (mostly Prochlorococcaceae). The only exception was OTU161, a Flavobacteriaceae that was the

only indicator of a single category, in this case In-MA (Supplementary Fig. 8). Importantly, the absence of a particular OTU can also be informative for a particular reef category. For example, 10 OTUs were diagnostic across all analysed categories with the exception of In-MA. OTUs affiliated with the Piscirickettsiaceae and unclassified Rhizobiales (plus OTUs in the OCS155, Halomonadaceae and Pelagibacteraceae) are additional examples where absence is diagnostic of the In-MA category.

## Discussion

This study extrapolated microbial community composition from all currently available datasets relevant to the GBR and explored environmental drivers of microbial variation using eReefs at the relevant spatial and temporal scales. This approach allowed us to summarize seasonal shifts in the microbial community of GBR waters along an inshore to offshore gradient. Our extrapolations putatively represent ~90% of all GBR reefs. Microbial communities in coral reef waters respond, at least partially deterministically, to environmental fluctuations and drivers[17,32,42]. Our analyses support this inference by showing that microbial communities within GBR surface waters strongly correlate with their surrounding prevailing conditions (e.g., nutrient dynamics, temperature), both at broad geographical scales (i.e. across reef categories), and at the level of temporal-seasonal dynamics. A trend of increasing terrestrial input near-shore resulted in reduced richness and diversity of bacterial communities in In-MA and In-Porites (inshore) reefs. Reduced microbial diversity was also evident at riverine stations (Supplementary Results, Supplementary Figs. 10-14 and Supplementary Tables 4–5). In contrast, stable oligotrophic oceanic conditions correlated with more diverse and rich bacterial communities for the Out-Digit and Out-Soft (outershelf) reefs (Figs. 2 and 3). Substantial changes in bacterial community structure were also evident across the GBR shelf (Fig. 4), primarily explained by nutrient dynamics, though a general gradient of decreasing temperature from inshore towards outershelf reefs also correlated with these community changes (Supplementary Fig. 6). These bacterial community patterns suggest that terrestrial and riverine influences impose selective processes under which a narrower array of microbial taxa can thrive[43,44].

Riverine outflows are known to impact the health of inshore reefs as they carry organic and inorganic nutrients of terrestrial origin (such as agricultural fertilizers) onto the reef systems[3,4]. Particulate matter is usually deposited a few kilometres from river mouths, though dissolved nutrients can reach distances >100 Km[45]. Terrestrial inputs are a major influence on environmental variation of GBR benthic and pelagic habitats[46,47] and observed patterns from this study indicate that water chemistry and nutrient gradients are also a crucial driver of microbial community change across spatial scales of GBR surface waters. This appears to be a generality of coastal oligotrophic waters. For a coastal environment in the Sargasso Sea, changes in microbial communities were observed from estuarine nearshore sites across the continental shelf to offshore oligotrophic waters, with seawater temperature and distance from shore (as proxy for gradients in productivity, terrestrial input and nutrients) identified as the strongest drivers of microbial community composition[42]. Changes in bacterioplankton community structure related to freshwater runoff have also been described for Hawaiian reefs[48].

Seasonal variation influenced the richness and structure of bacterial communities, though trends were not as strong as observed for cross-shelf spatial variation (see Figs. 3 and 4). Lower bacterial diversity was evident for the wet season in the Mid-Mixed reef category and seasonality influenced bacterial community structure for the inshore In-MA and In-Porites categories, as well as Mid-Mixed reefs (Supplementary Fig. 6). For these three reef categories, seasonality likely represents a response to the combined effects of temperature and nutrient dynamics, with a relative decrease in the availability of nutrients and a rise in seawater temperature during the wet season (see Fig. 2). For the In-Porites and Mid-Mixed reefs, there was a contribution of chlorophyll *a* and salinity to bacterial seasonal dynamics. Seasonal changes in community structure may be the result of event-related phytoplankton blooms, which are known to quickly assimilate and turnover available pools of dissolved inorganic nutrients in the GBR lagoon at the beginning of the warm wet season[49,50]. Seasonal influence on microbial diversity (lower in the wet season) and community structure was also confirmed for In-Porites reefs located under the influence of river plumes (see Supplementary Discussion). These seasonal patterns in microbial community dynamics, even at smaller spatial scales of river influence, are consistent with previous studies on the GBR[30,32]. Irrespective of their spatial and temporal drivers, temperature and nutrient availability likely interact to establish selective processes that determine the assemblage of prevalent bacterial groups for each reef category. Inshore reefs are typically light limited (high attenuation of light due to high particulate and dissolved nutrients) and richer in heterotrophic processes, whereas outershelf reefs are more nutrient limited, particularly nitrogen limited (low DIN), and richer in autotrophic processes[45,49,50].

The shallow-water pelagic microbiomes of the GBR were dominated by autotrophic cyanobacterial families Prochlorococcaceae and Synechococcaceae, as well as the oligotrophic Pelagibacteraceae (Alphaproteobacteria). These bacterial groups are highly abundant in the global ocean, with Pelagibacteraceae (formerly SAR11 clade) accounting for up to a third of all cells present in the oceans' surface waters[51]. Across the GBR, Pelagibacteraceae reached mean relative abundances >30% in outershelf (Out-Digit and Out-Soft) and midshelf (Mid-Mixed) reef waters. Prochlorococcaceae and Synechococcaceae represent the main photosynthetic bacteria in the ocean[52,53]. Prochlorococcaceae reached mean relative abundances of ~25% in outershelf reef waters (Out-Digit and Out-Soft), while Synechococcaceae populations were >30% in inshore reefs (In-MA and In-Porites). Metabolic properties of the dominant bacterial taxa likely influenced these observed distribution patterns across the GBR shelf.

Bacterial families OCS155, Cryomorphaceae, Flavobacteriaceae and Synechococcaceae, OM60, and Rhodobacteraceae positively correlated with increasing nutrient loads (DIC, DOC and NH4) and thus dominated inshore reefs. OM60 is an oligotrophic gammaproteobacterial family known to encompass diverse metabolisms including aerobic anoxygenic photosynthesis[54], and was previously identified as being more abundant in marine coastal zones than in open-ocean surface waters[55]. Rhodobacteraceae are often characterised as opportunistic microbes correlated with poor reef health[21] and are commonly enriched in diseased corals[56]. Both these lineages were found at higher relative abundances on In-MA than In-Porites reefs. The high abundance of the Bacteroidetes families Cryomorphaceae and Flavobacteriaceae in inshore reefs is consistent with previous studies. Bacteroidetes have been used as fecal indicators and can act as a reservoir of resistance genes for other more pathogenic bacterial strains[57]. A recent study has proposed that increased levels of Bacteroidetes in reef waters are indicative of enhanced macroalgal growth and the onset of microbialisation in coral reefs[58]. Proximity to river mouths has also been linked to increases in microbial taxa implicated in coral disease[21,30]. In contrast, Prochlorococcaceae showed increasing relative abundance with decreasing nutrients, being more abundant on

outershelf reefs likely as a result of their photoautotrophic metabolism. Rhodospirillaceae also showed a tendency to increase with decreasing nutrients, but their known diversified lifestyles (including diazotrophs, opportunistic pathogens, and varied genera able to grow in anaerobic or aerobic conditions[59]), likely support their constant relative abundances across reef categories. The increase in Pelagibacteraceae and SAR86, two abundant marine bacterial lineages that exhibit metabolic streamlining[60], is consistent with the oligotrophic conditions found in the GBR lagoon and Coral Sea. Nutrient levels measured in these habitats were the lowest found across all parameters and throughout all datasets included in our meta-analysis (see Fig. 2).

The population of indicators identified for each reef category includes particular OTUs within the families Rhodobacteraceae, Synechococcaceae, Cryomorphaceae, Flavobacteriaceae, Pelagibacteraceae and Halomonadaceae, among others. Counter-intuitively, the less rich and less diverse inshore communities yielded a larger number of microbial indicators than the taxa-rich and diverse outershelf reefs. This pattern strongly relates to the number of unique OTUs found in the different reef areas. The outershelf reefs were characterised by only 15 indicator OTUs from a restricted number of families and appeared to support a higher abundance of ubiquitously distributed bacterial taxa. In contrast, inshore environments harboured a greater relative abundance of autochthonous microbial taxa that represented indicators of inshore and potentially degraded systems. The absence of diagnostic microbial taxa for individual reef categories is a good illustration of the spatial continuity observed for free-living microbial communities among reef categories. This strongly contrasts with host-associated microbial data, where the specificity of bacterial taxa towards their host is usually high and indicator taxa are common[32].

The cyanobacterial families Prochlorococcaceae and Synechococcaceae show opposing correlations to many of the environmental parameters included in the analyses (Fig. 5), indicating they occupy complementary photoautotrophic microhabitats. Prochlorococcus is commonly reported from oligotrophic waters, due to its capacity to take up low levels of organic nitrogen, whereas Synechococcus becomes increasingly dominant in nutrient rich waters[53,61]. The Prochlorococcaceae:Synechococcaceae relative abundance ratio represents a potential indicator for the contribution of nutrient concentration in coral reef waters. These patterns have also been observed for pristine versus human-influenced reef atolls, with a four-fold increase in nitrogen and phosphate concentrations associated with Synechococcus dominance from 9–15% to 64–66% of the cyanobacterial population[12]. However, even within a particular bacterial genus or species, there can be different lineages (i.e. 16S rRNA sequence variants) that are associated with distinct environmental conditions[32,62]. For example, the Prochlorococcus group is comprised of various ecotypes that are phylogenetically and physiologically distinct and whose abundance distributions respond according to environmental gradients[63]. These patterns are often attributed to the partitioning of environmental resources and niche spaces among taxa[64]. An index could be established that categorizes this ratio into levels that broadly relate to the availability of nutrients/contribution of terrestrial run-off. Additional indices could be developed to monitor eutrophication of GBR waters, for instance by including particular lineages (e.g., Prochloroccocus or Synechococcus) with different substrate affinities. Comparison of broader trophic groups may also prove valuable indicators of ecosystem health and/or function. For example, levels of typical copiotrophs such as families OCS155, Flavobacteraceae, Cryomorphaceae and Rhodobacteraceae, could be modelled against levels of oligotrophs such as Pelagibacteraceae and SAR86 to generate a complementary index for eutrophication

(e.g. Haas et al.[26]). Typical opportunistic bacteria, such as those exhibiting virulence towards benthic organisms (e.g., the families Rhodospirillaceae, Rhodobacteraceae and Vibrionaceae), could also be used as indicators of reef health and/or degradation.

Analyses of GBR microbial communities with extensive environmental metadata identified bacterial taxa that are indicative of particular conditions on the reef, either because they contribute to the processes underlying reef health, or because they occur as a consequence of those underlying processes. The causality of these relationships is difficult to attribute directly, though a non-mutually exclusive alternative explanation could be that variability in benthic cover and associated availability of labile organic matter released by dominant benthic primary producers[65] causes changes in the microbial communities[26]. For instance, a reduced Prochlorococcaceae:Synechococcaceae ratio at In-MA reefs may be related to the increasing contribution of organic nutrients of macroalgal origin. In-MA reefs are potentially influenced by positive feedback loops through which microbial changes related to increased macroalgae cover promote more advanced states of macroalgae domination[26,66]. Although the identification of microbial taxa and functions that contribute to a functioning reef (or to disturbed reef states) is a major objective of this work, further analyses are required to identify potential microbial indicators of reef health, microbialization, degraded environments and ecological tipping points. The Flavobacteriaceae-affiliated OTU161 was the only indicator for a single reef category (i.e. In-MA) and may represent a microbial indicator of ecosystem degradation on the GBR. Further work should be undertaken to assess the robustness of this indicator by establishing comprehensive baselines and additional experimental validation. Future research should also resolve dynamic causal relationships between environmental parameters, microbial communities and underlying coral reef ecosystem health.

Robust mapping of microbial communities across the GBR requires extensive baselines across temporal periods and spatial scales that reflect its expanse of 2300 Km and almost 3000 individual reefs. Microbial baselines could be achieved through a series of microbial observatories spanning key habitats to complement current reef-monitoring efforts (e.g. AIMS LTMP). Establishment of parallel cross-shelf transects that capture all six reef communities previously identified[35], with frequent sampling to capture inter-seasonal patterns, is proposed. Extension of microbial data to the far northern section of the GBR is required to ascertain whether pelagic microbial communities in this region follow the patterns identified here for other regions of the GBR. Sampling effort should also be directed towards characterising pelagic microbial communities over the outershelf tabular and corymbose hard coral (Out-Tab) and the outershelf branching hard coral (Out-Digit) communities, for which we could not robustly predict a microbial community. However, we suggest that inshore reefs demand the most intensive monitoring program, as this is the area exposed to the broadest range of environmental variation, and the highest degree of uncertainty due to proximity to land and human activities affecting inshore reefs. These interactions drive higher spatial heterogeneity of bacterial communities in inshore reefs as compared to more homogeneous outershelf and open ocean sites[67].

Microorganisms represent the first responders to environmental change and may mitigate or exacerbate the impacts of disturbance for higher trophic levels[11]. Establishment of a GBR microbial observatory network would aid identification and validation of microbial indicators for environmental disturbance and facilitate early identification of ecosystem conditions leading to tipping-points in reef condition. Extending beyond taxa to also characterise the functional traits of these microorganisms would determine if changing community composition translates to

changes in biogeochemical cycling and/or other microbial processes important for ecosystem health[58]. Finally, it is crucial to generate global reef microbial data to understand the ubiquity of the patterns described here for the GBR.

## Methods

To estimate the compositional variation of microbial communities in GBR waters across different environments from inshore to outershelf reefs, we undertook a modelling exercise extrapolating the publically available microbial data (for a number of sites) into the larger extent of the GBR. The crucial step here was to ascertain the spatial representativeness of the available case study sites in relation to the wider GBR, which we did by modelling their environmental variation and comparing it to that of known GBR reef habitats as categorized in a recent study[35]. Our approach comprised: (1) deriving water chemistry data for sites with microbial data as well as for broad reef categories as monitored by the LTMP of the AIMS; (2) training a classification model that establishes a relationship between reef categories and prevailing environmental conditions; (3) predicting reef category for the microbial case study sites; and (4) summarizing microbial community data for broad reef categories across the wider GBR.

**Accumulated datasets for GBR microbial predictions**. The GBR represents an extensive reef system with large longitudinal temperature gradients and a seawater productivity gradient mostly defined by distance to shore[4]. The microbial datasets available as case studies for the GBR are spatially restricted, covering localised sites on Heron Island[40] and the Mackay region[31] in the southern GBR, the Burdekin region[32] in the central GBR, and the Tully region[30] in the northern GBR (see Fig. 1). Additional unpublished datasets (available through BioPlatforms Australia, BPA) were also incorporated into the meta-analysis, covering the Yongala lagoon site and several Coral Sea sites (Fig. 1) currently monitored by the Integrated Marine Observing System (IMOS) National Reference Station (NRS) Network[41]. In contrast, an extensive reef water quality program has been implemented for the GBR with the contextual data publicly available within the eReefs platform[68]. eReefs integrates a hydrodynamic model (predicting key environmental conditions such as temperature and salinity) and a biogeochemical model (water chemistry and ecological processes driving the water chemistry) to generate estimates with a high spatial and temporal resolution for a vast variety of environmental conditions across the GBR, from inshore reefs to the open ocean. Aggregations of such data are available over hourly, daily, monthly, annual or all-time periods (https://ereefs.org.au/).

To extend predictions of microbial community composition from the case study sites to the wider GBR, we use the coral community mapping of Mellin et al.[35] as a categorization framework. This classification splits reefs surveyed across the GBR as part of the LTMP of the AIMS[39] into six main reef benthic categories according to environmental predictors (such as distance to the barrier reef edge, seasonal range in seabed oxygen concentration and temperature, seasonal range in sea surface temperature, and percentage of carbonate sediments) (see Fig. 1). These reef benthic categories differ in their macroorganism indicator taxa and geographic position across the GBR shelf and are represented as: (1) outershelf soft coral communities (Out-Soft), (2), outershelf branching hard coral (Out-Digit), (3) outershelf tabular and corymbose hard coral (Out-Tab), (4) midshelf turf algae communities (Mid-Mixed), (5) inshore hard coral communities (In-Porites) and (6) inshore macroalgae communities (In-MA)[35].

To compare surface seawater conditions across sites for the available microbial case studies with those included in the LTMP program, environmental data available from the eReefs hydrodynamic and biogeochemical model (GBR1, https://research.csiro.au/ereefs/models/model-outputs/gbr1/) were extracted using the R package ereefs (https://github.com/AIMS/ereefs). Specifically, surface seawater temperature, salinity, total chlorophyll $a$, dissolved inorganic carbon (DIC), nitrogen (DIN) and phosphorus (DIP), ammonium (NH4), nitrate (NO3), dissolved organic carbon (DOC), nitrogen (DON) and phosphorus (DOP), total carbon (TC), nitrogen (TN) and phosphorus (TP), total suspended solids (TSS) and the vertical attenuation coefficient of light (Kd_490) were retrieved. For each of these 16 environmental variables, data spanning the period Jan 2015–Jan 2018, were extracted for each $1 \times 1$ Km grid cell matching the $n = 37$ sites that are part of the analysed case studies (further referred to as "microbial" sites) and $n = 109$ reference sites included in the GBR-wide analysis of Mellin et al.[35] (further referred to as "LTMP" sites) using an adapted R script (https://github.com/sammatthews990/eReefs_Fradeetal2019). Data for each site were then averaged for two seasons, wet and dry, as formally defined by the Australian Bureau of Meteorology (http://www.bom.gov.au/climate/glossary/seasons.shtml), i.e., warmer wet season defined as Dec-Feb, and cooler dry season as Jun-Aug. After assessing collinearity among the 16 environmental variables (module of correlation coefficient higher than 0.7, using function "ggpairs" in package GGally), non-collinear variables were included in a Linear Discriminant Analysis (LDA, function "lda" in package MASS[69]). LDA identifies the component axes that maximize the variance of the data, but additionally finds the axes that maximize the separation between multiple data classes. LDA was initially trained with the LTMP dataset to predict the reef category in the classification system of Mellin et al.[35] based on the

reduced set of environmental variables. LDA performance was evaluated using "leave-one-out" cross validation (LOOCV[70]), with function "table", and with Cohen's weighted Kappa for interrater agreement[71], via function "kappa2" in package irr, where disagreements were weighted according to their squared distance from perfect agreement. The trained LDA model was then used to predict reef categories for the reefs included in the microbial dataset using the prevailing environmental data as model input (using function "predict"). This allowed identification of the reef community classes of Mellin et al.[35] that show environmental variation similar to, or in the range of that occurring in the microbial sites, thus facilitating extrapolation of microbial community data to each of the reef benthic categories to predict putative community composition across the wider GBR.

**Summarizing microbial community data**. Microbial community data originating from the GBR case studies was obtained in the form of "species versus samples" tables, or tables of operational taxonomic units (OTUs) for which microbial taxonomy had previously been assigned using 16S rRNA phylogenetic marker genes. OTU tables quantified abundance of the 16S rRNA gene sequences affiliated with each microbial lineage across sampling sites, which provided an accepted proxy for relative abundances of distinct microbial taxa. Limitations of this metadata approach are that methods differed among case studies (see Supplementary Table 1), including: (i) different primer sets to amplify the 16S rRNA gene, (ii) different sequencing platforms, (iii) different reference databases to infer taxonomic affiliation of the 16S rRNA gene reads, (iv) different taxonomic resolution, and (v) different sampling depths, such that only the 2–5 m depth was used here, unless specified otherwise. These limitations highlight that caution is required when inferring trends across studies. Therefore we focus primarily on changes in microbial communities within each of the individual case studies rather than inferring responses across datasets (please see Supplementary Figs. 15–17, and Supplementary Discussion for further comments on putative limitations of this meta-analysis).

All OTUs derived from cellular plastids (mitochondria and chloroplast) were removed from the analyses and, because Archaea were only reported for the Tully region[30], our analyses are restricted to the domain Bacteria (unless specified). OTUs with one single occurrence (singletons) were removed to avoid including spurious data originating from sequencing errors. All data were rarefed for within-study comparisons (see Supplementary Table 1). For all data generated by BioPlatforms Australia (BPA) (Burdekin, Yongala and Coral Sea datasets), further documentation outlining the standard operating procedures for generating and processing sequencing amplicons is available online (https://data.bioplatforms.com/organization/pages/bpa-marine-microbes/methods). Otherwise, all preprocessing data analyses are detailed in the respective publications for each of the GBR regions: Tully[30], Burdekin[32], Mackay[31] and Heron Island[40].

All microbial data were transformed into relative abundance data and presented figures summarize the most abundant microbial taxa in each case study. Alpha-diversity estimators (richness, Shannon and Chao diversity) were calculated after rarefying (to 25,000 reads), and a mixed-effects Analysis of Variance (ANOVA; function "lmer" in lme4 package[72] and "rand" of lmerTest package[73]) applied to test for the effect of reef category and season (fixed factors) on each estimator (while using the original site as random effect). Venn diagrams were constructed with VennDiagram package[74] to depict the number of unique and shared OTUs. Beta-diversity statistics and visualizations were calculated from Bray-Curtis similarity matrices based on Hellinger-transformed abundance data to reduce influence of dominant lineages. Non-metric Multidimensional Scaling (NMDS—function "metaMDS" of vegan package[75]) limited to two dimensions was used to visualize the microbial community structure. Permutational Multivariate Analysis of Variance (PERMANOVA; function "adonis2" of vegan package[75]) was used to test (using 9,999 permutations) for differences in community structure between reef categories and seasons within each dataset (original site included as random effect). All beta-diversity statistics and follow-up analysis (see below) included only OTUs seen more than five times in at least 50% of the samples (dominant taxa). However, data obtained by 454 sequencing technology (Tully dataset) yielded a low number of OTUs and all OTUs were included in beta-diversity statistics and follow-up analysis.

**Drivers of microbial variation across the GBR**. To identify environmental drivers of microbial variation across GBR surface waters, environmental data available for the microbial case studies was again extracted from the eReefs hydrodynamic and biogeochemical model using the R package ereefs, with spatial and temporal resolution matching the available microbial datasets (and the place/time reported for their collections). All surface seawater data were extracted using $1 \times 1$ Km resolution except for those cases where only the $4 \times 4$ Km model could provide data, such as for the Tully region ($1 \times 1$ Km model only available from 2015 onwards). In all cases, data were averaged across the 3 days leading up to (and including) the actual sampling dates reported in the case studies. eReefs only models data obtained since 2011 (using $4 \times 4$ Km model), so microbiome-environment links could not be established for the Mackay region as this study[31] was conducted in 2009–2010. This precludes identification of the drivers of microbial variation for the Alongi et al.[31] dataset.

Environmental variation was visualized by Principal Component Analysis (PCA, function "prcomp" in package ggfortify[76]) of non-collinear variables (chosen as explained above for the GBR-wide dataset). Before PCA, abiotic variables were first checked for normality using a graphical check (and transformed if needed) and then scaled to zero mean and unit variance (z-scores standardization). To determine the contribution of abiotic parameters to the structure of microbial communities, a Bray-Curtis distance-based redundancy analysis (db-RDA) was applied on Hellinger-transformed relative abundance and the abovementioned non-collinear environmental metadata using functions "dbrda" and "capscale" of vegan package[75]. Samples with missing values for at least one of the environmental parameters were removed. A model selection tool based on the explained variance (function "ordiR2step") was used to select the environmental variables (constraints) significantly explaining variation in the microbial community (pin = 0.05, max perm 200), after which the significance of each constraint was confirmed with analysis of variance (ANOVA) for db-RDA (function "anova.cca" in the vegan package[75]). An ANOVA-like permutational test (function "permutest") for dbRDA was used to assess the significance of the full model. Explanatory value (in %) of significant constraints (e.g., environmental parameters, season and sampling date) was assessed with a Variation Partitioning Analysis of the vegan package[75]. The function "taxa.env.correlation" of microbiomeSeq package was used to compute (Spearman) correlation coefficients between environmental gradients and the relative abundance of the 20 most abundant OTUs (non-transformed data), for the environmental variables that were significant constraints of the community ("ordiR2step" above). The Benjamini-Hochberg procedure was used to calculate adjusted $p$-values for multiple comparisons.

Finally, to identify groups of microbes diagnostic of the different reef categories, an indicator value analysis (IndVal; indicspecies package[77]) was performed on relative abundance data (non-transformed). IndVal identifies microbial taxa based on their significant specificity and fidelity to particular treatments[78] and has been previously used in the coral microbiome literature[79].

**Statistics and reproducibility**. No experiments were performed. A high number of permutations was applied to deal with hypothesis-driven statistical tests. All meta-analyses of available microbial community composition and contextual environmental data were performed in R version 3.4.3[80] using the phyloseq package[81], and graphical outputs were generated with ggplot2 package[82].

**Reporting summary**. Further information on research design is available in the Nature Research Reporting Summary linked to this article.

## Data availability
All sequencing data obtained from BioPlatforms Australia (BPA), covering the Burdekin, Yongala and Coral Sea datasets, is available online: https://data.bioplatforms.com/organization/pages/bpa-marine-microbes/ Availability of all other sequencing data (reported in supplementary materials) is detailed in the respective publication for each of the GBR regions: Tully[30], Mackay[31] and Heron Island[40]. All environmental data used are available from the eReefs hydrodynamic and biogeochemical model (GBR1). Source data underlying the graphs and charts presented in the main figures are available as Supplementary Data.

## Code availability
An adapted R script[83] was developed to extract environmental data from eReefs spanning the period Jan 2015–Jan 2018 from each $1 \times 1$ Km grid cell matching the $n = 37$ microbial sites and $n = 109$ LTMP reference sites which is available here: https://github.com/sammatthews990/eReefs_Fradeetal2019

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

## Acknowledgements
We are grateful to Angus Thompson, Eric Lawrey and Barbara Robson from the Australian Institute of Marine Science for scientific advice on environmental variation across the GBR and the use of eReefs data; Florent Angly and the Australian Centre for Ecogenomics for access to sequencing data. This study was supported by funding from the Australian Government's National Environmental Science Program (NESP) Tropical Water Quality Hub. P.R.F. and E.A.S. were supported by the Portuguese Science and Technology Foundation (FCT) through fellowships SFRH/BDP/110285/2015, SFRH/BSAB/150485/2019 and UID/Multi/04326/2019 to Centre of Marine Sciences.

## Author contributions
P.R.F., D.G.B., N.S.W., K.W. and P.J.M. designed the study. P.R.F. analysed all environmental and microbial community data and produced a first draft of the manuscript. P.R.F., B.G. and S.A.M. produced all figures. S.A.M. and C.M. developed code to facilitate extraction of eReefs data with spatial and temporal resolution. P.R.F., B.G., S.A.M., C.M., E.A.S., K.W., P.J.M., N.S.W., and D.G.B. edited through several versions of the manuscript and contributed to its final version.

## Competing interests
The authors declare no competing interests.
