## [Peer Review File · Communications Biology]

Reviewers' comments:

Reviewer #1 (Remarks to the Author):

This manuscript demonstrated the bacterioplankton community dynamics across surface-waters of the Great Barrier Reef (GBR) through a meta-analysis. A big dataset combining microbial communities with environmental data from the eReefs platform is the base of this study. The advantage of this study is that the microbial data were collected at adequate spatial and temporal resolution in the GBR, and supported by a comprehensive suite of contextual parameters. The main result indicated that the nutrient dynamics and temperature could explain 41.6% of the inter-seasonal and cross-shelf variation in bacterial community assemblages. It improves the understanding of microbial community dynamics in the context of the ecosystem functioning in the GBR. Such a wide-ranging synthesis is essential to scale-up management of natural ecosystems under threatening. Overall, this manuscript is well organized with a valuable dataset. Some restrictions about data quality and interpretation have also been declared.

Here I have some comments and suggestions.

1. Possible primer bias and sequencing platforms discrepancy are the weak points of meta-analysis, which may limit the following interpretation considering the whole dataset. It's necessary to state the data limitation in the main content to avoid any confusion or over expectation. It would be even better if the authors can evaluate the effect of original data bias.
2. Please clearly state how to get the diagnostic microbial community (fig.5) for each reef category. I wonder the validity of assembling sequencing data from different sites to obtain the representative microbial community for each reef category (I may get lost in somewhere). Besides, I suggest using the Venn diagram (by Family or OUT level) to clarify the similarity and difference of microbial communities among these reef categories.
3. Environmental drivers of microbial community changes have been well characterized among different reef categories with anticipation because microorganisms rely on environmental conditions and water chemistry deeply, and so does the coral reef. However, it is not clear whether these environmental drivers or microbial indicators can actively influence ecosystem health and function regarding the coral reef. It is possible that the environmental drivers can only drive the change of microbial communities but not coral reef ecology. Therefore, I expect to see more discussion about the relationship between environmental drivers and reef categories, which may give us a hint about the real drivers to the coral health.
4. Most microorganisms detected in this study are also abundant in coastal and marine environments. Thus, only the diagnostic microbial community of each reef category in the GBR can represent a specific microbial ecosystem associated with a particular type of coral reef system. In my opinion, the community structure of microbes might be more important than the existence of specific Families or OTUs. A comparison between microbial communities from other sites with a similar type of coral reefs can demonstrate the uniqueness or generalization of the diagnostic microbial community in the GBR. I think it is critical to show that the association between the diagnostic microbial community and the reef category is unique in the GBR. Otherwise, the importance of this study might decrease.
5. A reference list is needed for supplementary.

Reviewer #2 (Remarks to the Author):

Review of Frade et al. 2020 Communications Biology "Spatial patterns of microbial communities across surface waters of the Great Barrier Reef"

Frade et al. present a compelling meta-analysis of the bacterioplankton communities of coral reefs across the Eastern Australian Great Barrier Reef. Many of the patterns they distill are widely consistent with previous work, and as such this meta-analysis is useful in a general sense to provide a basis for general reef patterns. Reiterating that *Synechococcus* and *Prochlorococcus* are, respectively, nearshore and offshore dominant, is not novel but nice to see in a meta-analysis, and many of the organisms enriched nearshore (Flavo, Cryo, Rhodo, OM60 and OCS155) are clades that many have reported associated with reef systems. I think this paper is useful and I hope that the authors are able to improve it for publication.

Where this manuscript struggles is in the extensive and complex modelling exercises. They are unclear and potentially problematic, and I encourage the authors to carefully address my recommendations below to improve the presentation. There are many areas where things are wrong, or just not clear, and there is really not much effort put into distilling the data into clear patterns, instead simply regurgitating a lot of complexity. This is fine, as long as done correctly, but I see many areas for improvement.

Please don't submit manuscripts for review without page numbers. It is a headache for reviewers, and then a headache for authors trying to figure out what items reviewers are referring to. Frankly this is the journal's shortcoming even allowing you to do this, and I have pointed this out to the editor, but I also want to remind you how important it is to appreciate the work of reviewing manuscripts and making the process as simple as possible.

Abstract: Pro vs. Syn as a ratio "representing a potential indicator of nutrient ENRICHMENT" is not supported.

Also inappropriate to describe "water quality niches" as these are just biogeochemical niches.

Generally speaking, I would scan the manuscript for "water quality" and "nutrient enrichment" and be very careful how and when you use those terms. The first doesn't mean anything, so be clear what you mean. The second implies active enrichment, not just concentration gradients, and again you are often implying something you haven't addressed. These organisms track gradients (as you yourself describe quite nicely in the discussion) and currently don't tell us anything about active enrichment.

Page 10: Because you have autocorrelation between datasets (done in different years, with different methods) and Mellin reef types (modeled with LDA from eReef models of water chemistry) you are stuck with the mandate that you include "study" or "site" in your PERMANOVA models, partitioning variance accordingly. I believe that some algorithms can use random effects in PERMANOVA, and this would be appropriate for the "study" term so that you can resolve the other effects of Reef Category and Season (Assuming I am reading Page 11 correctly...it is not crystal clear how you structured your adonis models).

Page 10-11: The alpha diversity analyses need statistics, incorporating season and reef category into a 2-way anova (and see above, adding a random effect with lmer of "study" or "site")

Page 25: It is not clear what you did, or why you did it. It sounds as though you wanted to be able to predict what the benthic cover categories were at the sites where you had microbial data, so you...

1) derived water chemistry for microbial data sites and LTMP sites (specifically which variables...you just state "temperature, salinity, water quality (nutrients and suspended sediments) and water chemistry"

2) "summarized per season" - what does this mean? Are we now working with four datasets? Means?

3) ran PCA separately (using z-scores of those variables) on microbial and LTMP sites. z-score normalization (please call it standardization) first requires that abiotic data all be normally distributed (so that you can calculate a mean and sd of the distributions to derive z-scores). Please be explicit that you did this, or rebuild the model, as PCA and z-scoring are both sensitive to normal distributions (ie Gaussian assumptions). Data don't have to be perfectly (or statistically) normal, but can't be strongly skewed.

4) Trained LDA with ("reduced set of environmental variables" from PCA) of the variables on the LTMP dataset to predict Mellin categories. What were those variables? Were they a handful of PCA eigenvalues? Say the first four components? Which? And why not just put all of the variables into LDA? You are already avoiding covariation using ggpairs...

5) Used trained LDA to predict the Mellin categories for the microbial sites from the water chemistry data.

So, in summary, you modeled the benthic cover of the microbial sites using grid-interpolated (modeled via GBR1/eReef) water chemistry data. This is very very hard to tease out of your methods. It is also tenuous - you are modeling the water chemistry, then trying to train a model to predict the benthic cover category, and somewhere in the middle you run a dimensional reduction algorithm (PCA) which does not have a clear purpose other than to further "model" the data.

Finally, it is not clear where this was used. Was this just used in some kind of classification of the microbial sites? Were these classes used in the db-RDA analyses as environmental covariates? Explain what the goal of this whole exercise was.

Page 27: "after rarefying" - it would be nice to know to what read depth you rarefied in each region.

Page 27: NMDS doesn't "assess dissimilarity". It is a visualization and dimensional reduction tool.

Page 27: Why use both ANOSIM and PERMANOVA? Pick and justify one.

Page 8: "which 16 were considered informative for microbial communities based on current literature" - Figure S1 lists 16, but doesn't establish how they were chosen as being "informative for microbial communities". Figure 2 does not hint at 16 variables. Make this entire sentence much more crystal clear, as it also doesn't mesh with your modeling exercise with LDA.

Page 8: "Results on environmental variation across the GBR are detailed in the supplementary materials" means nothing. What are you trying to say?

Figure 2A does not explain the variables on the biplot arrows - they are just codes. These should be interpretable names of variables and explained in the legend and text.

Figure 2B can be moved to a supplement. It doesn't help inform the reader of anything but the accuracy of the LDA model.

Figure 2c could be replaced by color coding the microbial sampling sites according to their modeled Mellin categories on the map in Figure 1. Then in Figure 1 make it abundantly clear that the microbial samples are modeled while the LTMP sites are measured. This will also be more informative than the proportions in Figure 2c.

Figure 2d is simply uninformative, plus it is showing a PCA of modeled information which is misleading.

I recommend that Figure 2 be moved to three supplements (a, b, and c+d) and that you provide means for the various water chemistry parameters in the Mellin categories in both the measured (LTMP) and LDA-modeled (microbial) datasets. This will be much more digestible than the PCA plots.

Page 12: This discussion is great, and is consistent with many other studies of microbes across coral reefs. For example, the work of Yeo et al 2013 Plos ONE, Nelson et al 2011 ISMEJ, McCliment et al 2011 ISMEJ, all track these gradients and discuss the common organisms found nearshore.

Page 13: Dimension reduction is not shown in Figure S5. Are you saying that the PCAs in Figure S6 (and maybe Figure 2) only use 8 variables (or less in Figure 2) because you have done some kind of dimensional reduction (presumably with ggpairs?). This should be clear and easy to follow.

Page 13: I don't follow the sentence "Replacing environmental variables by reef category and season explained 32.5% and 7.1% of variation, respectively."

I find Page 14-15 to be a useful discussion and Figure 4 does not do it justice. I recommend that Figure 4A be amended with some simpler graphical depictions of some of the patterns discussed in this paragraph, as these are the crux of the paper and are lost in Fig 4. If the authors can use clustering or other methods to synthetically represent and distill some of the patterns great, otherwise just simple wet/dry correlations between environment and selected taxonomic groups would be insightful.

Figure 5b: the pie charts should be sized (or annotated with a number) according to the summed relative abundance of indicator organisms at each site. This will help clarify that indicators of offshore sites (Pro) are very abundant in those sites, while indicators of nearshore sites tend to be rarer organisms.

Also, for the sake of the sanity of coral microbiology, please annotate the figure itself to clarify that these are relative abundances of indicator organisms only, and that this does not represent the bacterioplankton communities.

Figure 4b seems like it should be earlier in the paper, and it is not clear if it is from LTMP or the modeled microbial sites. See my earlier comments requesting something like this. What purpose does it serve here?

Page 21: replace "nutrient enrichment" with "nutrient concentration"

Page 21: replace "driving" with "associated with"

Page 21: "Additional indices could be developed to monitor eutrophication of GBR waters, for instance by including particular lineages (of *Prochlorococcus*) with different substrate affinities." Because Pro is not strongly associated with nearshore habitats, I would argue that this would be better suited to exploring Syn and/or other microbial taxa for "nutrient-affiliated ecotypes" or whatever. Just because

Pro has been studied in this capacity does not make it a good indicator for terrestrial nutrient inputs.

	Comments from the Editor	Response from authors
	At this stage, we ask that you please show that the association between diagnostic microbial communities and reef category is unique to GBR, as requested by reviewer #1. We also ask that you please address all the concerns raised by reviewer #2 related to the statistical analyses and modeling. Lastly, we also encourage that you incorporate all relevant methodological details and that you edit the manuscript for readability as the referees were unclear about the rationale behind some of the analysis.	We are very grateful for the opportunity to provide a revised version of our manuscript and have addressed all comments provided by the reviewers in our revised document. With respect to this general comment by the Editor, we refer to our email correspondence in February 2020. As explained then, and elaborated on below in our response to comment #4 of reviewer #1, due to a current lack of global data it is impractical to “show that the association between diagnostic microbial communities and reef category is unique to GBR”. Re-running our analysis for broader environments is technically impossible due to limitations in the availability of global data for microbial communities in reef systems, and particularly due to the lack of comparability among existing datasets. However, we have addressed this comment by strengthening and highlighting how our findings compare to other reef systems, now discussing “the uniqueness or generalization of the diagnostic microbial community in the GBR” as requested by reviewer #1. Please see our response below.

	Comments from reviewer #1	Response from authors
	This manuscript demonstrated the bacterioplankton community dynamics across surface-waters of the Great Barrier Reef (GBR) through a meta-analysis. A big dataset combining microbial communities with environmental data from the eReefs platform is the base of this study. The advantage of this study is that the microbial data were collected at adequate spatial and temporal resolution in the GBR, and supported by a comprehensive suite of contextual parameters. The main result indicated that the nutrient dynamics and temperature could explain 41.6% of the inter-seasonal and cross-shelf variation in bacterial community assemblages. It improves the understanding of microbial community dynamics in the context of the ecosystem functioning in the GBR. Such a wide-ranging synthesis is essential to scale-up management of natural ecosystems under threatening. Overall, this manuscript is well organized with a valuable dataset. Some restrictions about data quality and interpretation have also been declared.	We thank the reviewer for this favourable review. Please see our detailed responses below.
1	Here I have some comments and suggestions. 1. Possible primer bias and sequencing platforms discrepancy are the weak points of meta-analysis, which may limit the following interpretation considering the whole dataset. It's necessary to state the data limitation in the main content to avoid any confusion or over expectation. It would be even better if the authors can evaluate the effect of original data bias.	We appreciate the reviewer's comment and agree that there are indeed some limitations with such meta-analysis processes. We now highlight these limitations and have included reference to different sequencing platforms, which we missed in the previous version. Please see lines 904-908: “Limitations of this metadata approach are that methods differed among case studies (see Supplementary Table 1), including: i) different primer sets to amplify the 16S rRNA gene, ii) different sequencing platforms, iii) different reference databases to infer taxonomic affiliation of the 16S rRNA gene reads, iv) different taxonomic resolution, and v) different sampling depths...” We note, however, that the main patterns were all extracted from datasets for which the same methodologies were used, as they were all generated by the consortium BioPlatforms Australia (see lines 302-305). As stated in the Supplementary Material, the datasets originating from the Burdekin region, including the Yongala lagoon site, plus the Coral Sea

		dataset were obtained with the same primer set and processed through the same analysis pipeline of BioPlatforms Australia (please see Supplementary Table 1). Therefore we do not see the need to evaluate the effect of data bias. We have however added another cautionary note to the Materials and Methods section. Please see lines 910-913: “Therefore we focus primarily on changes in microbial communities within each of the individual case studies rather than inferring responses across datasets (please see Supplementary Discussion for further comments on putative limitations of this meta-analysis).”
2	2. Please clearly state how to get the diagnostic microbial community (fig.5) for each reef category. I wonder the validity of assembling sequencing data from different sites to obtain the representative microbial community for each reef category (I may get lost in somewhere). Besides, I suggest using the Venn diagram (by Family or OUT level) to clarify the similarity and difference of microbial communities among these reef categories.	To clarify our methodological approach, we have added additional detail as requested (also following advice from reviewer 2) (please see lines 789-800). In addition, we added details of the IndVal analysis to identify the diagnostic microbial community, as well as citing previous literature that includes detailed descriptions of the individual methods. Please see lines 1002-1006: “Finally, to identify groups of microbes diagnostic of the different reef categories, an indicator value analysis (IndVal; indicpecies package⁷⁷) was performed on relative abundance data (non-transformed). IndVal identifies microbial taxa based on their significant specificity and fidelity to particular treatments⁷⁸ and has been previously used in the coral microbiome literature⁷⁹.” We greatly appreciate the suggestion of adding a Venn diagram as it nicely complements information provided in Fig. 6 (Fig. 5 previously). A Venn diagram is now presented in Fig. 3, so it comes together with the other information on alpha-diversity. Similarities and differences of microbial communities among reef categories as retrieved from the Venn diagram are now presented in the text in lines 312-319: “A substantial number of OTUs were unique to a particular reef category (Fig. 3b), with the midshelf reef (Mid-Mixed) having the highest proportion of unique OTUs (63% of all OTUs found in that category), followed by inshore macroalgae communities (In-MA; 51%) and outershelf soft coral communities (Out-Soft; 33%). Outershelf branching hard coral (Out-Digit, only 13% unique OTUs) and inshore hard coral communities (In-Porites, only 9% unique OTUs) shared most OTUs with the

		other category in their respective reef shelf region, i.e., Out-Soft (with which Out-Digit shared 43% of all OTUs) and In-MA (with which In-Porites shared 54% of all OTUs).” Importantly, in the Venn diagram microbial taxa allocated to a specific sampling group (here, reef category) only have to present in a single sample for this taxa to be quantified as unique/shared. In contrast, the more stringent IndVal approach also considers fidelity (microbial taxa present in most samples of that particular group) in order for that microbial taxa to be flagged as diagnostic for that group.
3	3. Environmental drivers of microbial community changes have been well characterized among different reef categories with anticipation because microorganisms rely on environmental conditions and water chemistry deeply, and so does the coral reef. However, it is not clear whether these environmental drivers or microbial indicators can actively influence ecosystem health and function regarding the coral reef. It is possible that the environmental drivers can only drive the change of microbial communities but not coral reef ecology. Therefore, I expect to see more discussion about the relationship between environmental drivers and reef categories, which may give us a hint about the real drivers to the coral health.	The reviewer raises an excellent point regarding causality of patterns and points towards the wide range of ecological factors that could be discussed in greater detail. However, while it is crucial to establish causal links between environmental conditions and reef ecosystem health, it was not the primary purpose of this study. Our work did not assess coral health or coral-associated microbial communities, but rather focussed on bacterioplankton communities. The relationships between environmental drivers and reef habitat and function are the focus of other studies, such as Mellin et al. (2019) which defined the benthic reef categories used in the current study. Many previous studies have focused on the role of particular environmental drivers, such as temperature, sedimentation, terrestrial run-off, levels of eutrophication, in the response of coral reef communities. Here, we can only briefly summarise that literature. A part of the discussion (see lines 737-755) already included some of these ideas, however, we have now added a specific sentence to state that further research is required to disentangle the relationships between environmental drivers, microbial communities and the ecology of the reef. Please see lines 756-757: “Future research should also resolve dynamic causal relationships between environmental parameters, microbial communities and underlying coral reef ecosystem health.”
4	4. Most microorganisms detected in this study are also abundant in coastal and marine environments. Thus, only the diagnostic microbial community of each reef category in the GBR can represent a specific microbial	If we understand this point correctly, the reviewer argues that the association between diagnostic microbes and reef category is particularly relevant when shown to be unique to the GBR. We respectfully disagree and would argue that the fact that these findings

ecosystem associated with a particular type of coral reef system. In my opinion, the community structure of microbes might be more important than the existence of specific Families or OTUs. A comparison between microbial communities from other sites with a similar type of coral reefs can demonstrate the uniqueness or generalization of the diagnostic microbial community in the GBR. I think it is critical to show that the association between the diagnostic microbial community and the reef category is unique in the GBR. Otherwise, the importance of this study might decrease.	could be true for other reefs only strengthens the impact of this work; that microbial communities are structured according to habitats within reef systems, and that these patterns are the result of ecological and microbiological patterns that are likely to be common for different reef systems. Regardless, we believe that redoing the entire analysis for broader environments is technically impossible due to limitations in the available global data for microbial communities of reef systems, and particularly due to the lack of comparability among different datasets. However, we address this comment by strengthening and highlighting how our findings compare to other reef systems, so we can demonstrate “the uniqueness or generalization of the diagnostic microbial community in the GBR” as requested. We have therefore compared the patterns found here to those found in other reef locations. We have also added additional information from relevant literature. Please see lines 610-612: “Changes in bacterioplankton community structure related to freshwater runoff have also been described for Hawaiian reefs⁴⁸.”; lines 672-674: “A recent study has proposed that increased levels of Bacteroidetes in reef waters are indicative of enhanced macroalgal growth and the onset of microbialisation in coral reefs⁵⁸.”; and lines 773-775: “These interactions drive higher spatial heterogeneity of bacterial communities in inshore reefs as compared to more homogeneous outershelf and open ocean sites⁶⁷.” In response to the comment that “the community structure of microbes might be more important than the existence of specific Families or OTUs”, we would like to clarify that our approach did not neglect community structure. Our approach was to show differences in community structure of microbes between the different reef categories (i.e., community changes across ecological gradients), and then provide evidence and examples for specific members of the seawater microbiome that are particularly important and diagnostic (i.e., so called bioindicators). This is a combined approach that will be valuable to apply to other reefs systems, but unfortunately there is
---	---

		currently a lack of global data to enable broader comparisons. Summarizing microbial communities for the GBR reef system is already a considerable challenge considering the absence of microbial baselines. We also reiterate that our comparisons were established based on a few datasets which were generated using the same collection, amplification and sequencing protocols. Our findings from this analysis are highly relevant as they highlight the need to 1) collect global data and 2) investigate how these compare across different biogeographies. We have added these points to the conclusion of the manuscript in lines 784-785: “Finally, it is crucial to generate global reef microbial data to understand the ubiquity of the patterns described here for the GBR.”
5	5. A reference list is needed for supplementary	This was an omission in the original version and we have now added a reference list to the resubmitted SOM.

	Comments from reviewer #2	Response from authors
1	Frade et al. present a compelling meta-analysis of the bacterioplankton communities of coral reefs across the Eastern Australian Great Barrier Reef. Many of the patterns they distill are widely consistent with previous work, and as such this meta-analysis is useful in a general sense to provide a basis for general reef patterns. Reiterating that Synechococcus and Prochlorococcus are, respectively, nearshore and offshore dominant, is not novel but nice to see in a meta-analysis, and many of the organisms enriched nearshore (Flavo, Cryo, Rhodo, OM60 and OCS155) are clades that many have reported associated with reef systems. I think this paper is useful and I hope that the authors are able to improve it for publication. Where this manuscript struggles is in the extensive and complex modelling exercises. They are unclear and potentially problematic, and I encourage the authors to carefully address my recommendations below to improve the presentation. There are many areas where things are wrong, or just not clear, and there is really not much effort put into distilling the data into clear patterns, instead simply regurgitating a lot of complexity. This is fine, as long as done correctly, but I see many areas for improvement.	We are immensely grateful to the reviewer for their supportive and highly constructive review, which has considerably improved our manuscript. We have incorporated all recommendations within the revised manuscript and address each specific point below.
2	Please don't submit manuscripts for review without page numbers. It is a headache for reviewers, and then a headache for authors trying to figure out what items reviewers are referring to. Frankly this is the journal's shortcoming even allowing you to do this, and I have pointed this out to the editor, but I also want to remind you how important it is to appreciate the work of reviewing manuscripts and making the process as simple as possible.	We believe the reviewer is referring to line numbers here, as the submitted manuscript included page numbers. We agree that reviewing manuscripts without line (or page) numbers can be challenging, and this recommendation should be incorporated in the journal's guidelines. We are pleased to provide a revised version with line (and page) numbers.
3	Abstract: Pro vs. Syn as a ratio "representing a potential indicator of	The term "nutrient enrichment has been replaced with "nutrient levels".

	nutrient ENRICHMENT” is not supported.	Please see lines 47-50: “Cyanobacteria from the Prochlorococcaceae and Synechococcaceae families occupy complementary cross-shelf biogeochemical niches, with their relative abundance ratios representing a potential indicator of nutrient levels at GBR sites.”
4	Also inappropriate to describe “water quality niches” as these are just biogeochemical niches.	This has been amended as suggested. Please see response to comment #3.
5	Generally speaking, I would scan the manuscript for “water quality” and “nutrient enrichment” and be very careful how and when you use those terms. The first doesn’t mean anything, so be clear what you mean. The second implies active enrichment, not just concentration gradients, and again you are often implying something you haven’t addressed. These organisms track gradients (as you yourself describe quite nicely in the discussion) and currently don’t tell us anything about active enrichment.	We agree that nutrient enrichment could be interpreted as active enrichment and have therefore corrected this term throughout the manuscript, as requested. Please see response to comment #3 and lines 706-707: “The Prochlorococcaceae:Synechococcaceae relative abundance ratio represents a potential indicator for the contribution of nutrient concentration in coral reef waters.” Our references to water quality were guided by the literature we cite that use that term (also frequently in the title). In instances where it was possible to do so, we have modified our terminology to be more specific about the component of water quality under investigation. Please see lines 80-83: “Shifts in the compositional and functional diversity of both coral-associated^{15,16} and free-living planktonic^{12,17} microbial communities have been linked to varying levels of anthropogenic impact, including changes in seawater nutrient levels¹⁸.”; lines 110-111: “Shifts in free-living microbial lineages in response to seawater nutrient gradients and benthic composition in reef systems have previously been reported.”; lines 134-137: “In the northern GBR (Tully River region), microbial communities in proximity to reefs followed seasonal dynamics and responded to riverine inputs, with rainfall, water quality (i.e., nutrient, organic compounds and herbicides), salinity and temperature implicated as drivers of bacterial community composition³⁰.”; lines 602-605: “Terrestrial inputs are a major influence on environmental variation of GBR benthic and pelagic habitats^{46,47} and observed patterns from this study indicate that water chemistry and

		nutrient gradients are also a crucial driver of microbial community change across spatial scales of GBR surface waters.”; lines 803-804: “The GBR represents an extensive reef system with large longitudinal temperature gradients and a seawater productivity gradient mostly defined by distance to shore⁴.”; and lines 815-817: “eReefs integrates a hydrodynamic model (predicting key environmental conditions such as temperature and salinity) and a biogeochemical model (water chemistry and ecological processes driving the water chemistry)...”.
6	Page 10: Because you have autocorrelation between datasets (done in different years, with different methods) and Mellin reef types (modeled with LDA from eReef models of water chemistry) you are stuck with the mandate that you include “study” or “site” in your PERMANOVA models, partitioning variance accordingly. I believe that some algorithms can use random effects in PERMANOVA, and this would be appropriate for the “study” term so that you can resolve the other effects of Reef Category and Season (Assuming I am reading Page 11 correctly...it is not crystal clear how you structured your adonis models).	Although the datasets used to investigate microbial communities reported in the main text were all obtained with identical methods (see lines 294-297 and 302-305), we realise that there are different sites involved. We also agree that there is a degree of autocorrelation between sites and Mellin reef types and have therefore followed the reviewers recommendation to include site as a random effect in the PERMANOVA models (adonis2). Site was used as a nested factor (within reef category) in the model, and permutations were limited to only those samples with the same levels of Reef Category and Site, so that each level is implicitly treated as having a disparate error structure (i.e., random effect). When running this new PERMANOVA model, we obtained significances similar to those found previously, for the overall model, as well as for the main factors (reef category and season), which we now report. Please see Supplementary Table 3 and lines 942-945: “Permutational Multivariate Analysis of Variance (PERMANOVA; function “adonis2” of vegan package⁷⁵) was used to test (using 9,999 permutations) for differences in community structure between reef categories and seasons within each dataset (original site included as random effect).” And lines 332-334: “(…)full model PERMANOVA with interaction, pseudo $F_{(9, 59)} = 14.04$, $p < 0.001$; see Supplementary Table 3).”
7	Page 10-11: The alpha diversity analyses need statistics, incorporating season and reef category into a 2-way anova (and see above, adding a random effect	These statistics were presented (see lines 305-307), but we have now updated our results after using “site” as random effect using functions lmer, anova and rand of the lme4 and lmerTest packages.

	with lmer of “study” or “site”	Please see methodological description in lines 933-936: “(…) and a mixed-effects Analysis of Variance (ANOVA; function “lmer” of lme4 package⁷² and “rand” of lmerTest package⁷³) applied to test for the effect of reef category and season (fixed factors) on each estimator (while using the original site as random effect).”; and lines 305-308: “Alpha diversity (observed richness; Fig. 3a) varied significantly among reef categories and differences were not homogeneous across seasons (significant interaction; $F_{(2,59)} = 14.43$, $p < 0.001$; see Supplementary Table 2 and Supplementary Fig. 4 for further results).”
8	Page 25: It is not clear what you did, or why you did it. It sounds as though you wanted to be able to predict what the benthic cover categories were at the sites where you had microbial data, so you... 1) derived water chemistry for microbial data sites and LTMP sites (specifically which variables...you just state “temperature, salinity, water quality (nutrients and suspended sediments) and water chemistry” 2) “summarized per season” - what does this mean? Are we now working with four datasets? Means? 3) ran PCA separately (using z-scores of those variables) on microbial and LTMP sites. z-score	We appreciate the reviewer comments, which we address sequentially. The reviewer correctly identified that our approach was undertaken because we “wanted to be able to predict what the benthic cover categories were at the sites where you (we) had microbial data”, although we realise our approach lacked necessary detail in the methodological description. 1) Correct. We have now added a detailed list of parameters used. Please see lines 842-846: “Specifically, surface seawater temperature, salinity, total chlorophyll a, dissolved inorganic carbon (DIC), nitrogen (DIN) and phosphorus (DIP), ammonium (NH₄), nitrate (NO₃), dissolved organic carbon (DOC), nitrogen (DON) and phosphorus (DOP), total carbon (TC), nitrogen (TN) and phosphorus (TP), total suspended solids (TSS) and the vertical attenuation coefficient of light (K_{d_490}) were retrieved.” 2) We averaged per season. For simplicity we use only the wet and dry seasons, as is now described in the methods. The tropical GBR regions that the datasets are derived from have two clearly defined seasons, i.e. a hotter and wetter summer season and a cooler and drier winter season (as formally defined by the Australian Bureau of Meteorology). Please see lines 851-854: “Data for each site was then averaged for two seasons, wet and dry, as formally defined by the Australian Bureau of Meteorology (http://www.bom.gov.au/climate/glossary/seasons.shtml), i.e. warmer wet season defined as Dec-Feb, and cooler dry season as Jun-Aug).” 3) Correct. We did this to understand if the underlying structure of the two datasets was similar. Following this and further reviewer

	normalization (please call it standardization) first requires that abiotic data all be normally distributed (so that you can calculate a mean and sd of the distributions to derive z-scores). Please be explicit that you did this, or rebuild the model, as PCA and z-scoring are both sensitive to normal distributions (ie Gaussian assumptions). Data don't have to be perfectly (or statistically) normal, but can't be strongly skewed. 4) Trained LDA with ("reduced set of environmental variables" from PCA) of the variables on the LTMP dataset to predict Mellin categories. What were those variables? Were they a handful of PCA eigenvalues? Say the first four components? Which? And why not just put all of the variables into LDA? You are already avoiding covariation using ggpairs... 5) Used trained LDA to predict the Mellin categories for the microbial sites from the water chemistry data.	comments below, we have opted to remove these PCAs from the main manuscript, as they indeed represented an unnecessary middle step. As such, there was no need for z-score normalization. 4) Correct. We used all non-collinear variables for the LDA (6 variables after excluding collinear variables from ggpairs). We did not use PCA eigenvalues here and PCAs are now removed from the manuscript (see above). We now clarify which variables were used and why, and have added all non-collinear variables into the LDA. Please see lines 854-857: "After assessing collinearity among the 16 environmental variables (module of correlation coefficient higher than 0.7, using function "ggpairs" in package GGally), non-collinear variables were included in a Linear Discriminant Analysis (LDA, function "lda" in package MASS⁶⁹)." 5) Correct. Our revisions to the methodology section now clearly describe this approach.
9	So, in summary, you modeled the benthic cover of the microbial sites using grid-interpolated (modeled via GBR1/eReef) water chemistry data. This is very very hard to tease out of your methods. It is also tenuous - you are modeling the water chemistry, then trying to train a model to predict the benthic cover category, and somewhere in the middle you run a dimensional reduction algorithm (PCA) which does not have a clear purpose other than to further "model" the data.	Correct. We "modelled the benthic cover of the microbial sites using grid-interpolated (modelled via GBR1/eReef) water chemistry data". We agree that the original approach was sub-optimal and have now removed the PCA analyses. We recognize that our approach is not consistent with most publications, but we also note the challenges inherent in the available datasets which necessitated some degree of modelling and assumptions. We rely on the correlations (ggpairs) to reduce dimensions in the dataset (as described in lines 854-857; see above). We have also elaborated on our general approach in the beginning of the methods section to improve clarity. Please see lines 789-800: "To estimate the compositional variation of microbial communities in GBR waters across different environments from inshore to outershelf reefs, we undertook a modelling exercise extrapolating the publically available microbial data (for a number of sites) into the larger extent of the GBR. The crucial step here was to ascertain the spatial representativeness of the

		available case study sites in relation to the wider GBR, which we did by modelling their environmental variation and comparing it to that of known GBR reef habitats as categorized in a recent study ³⁵ . Our approach comprised: 1) deriving water chemistry data for sites with microbial data as well as for broad reef categories as monitored by the Long Term Monitoring Program (LTMP) of the Australian Institute of Marine Science (AIMS); 2) training a classification model that establishes a relationship between reef categories and prevailing environmental conditions; 3) predicting reef category for the microbial case study sites; and 4) summarizing microbial community data for broad reef categories across the wider GBR.”
10	Finally, it is not clear where this was used. Was this just used in some kind of classification of the microbial sites? Were these classes used in the db-RDA analyses as environmental covariates? Explain what the goal of this whole exercise was.	The goal of our study was to predict benthic cover categories at sites for which we had available microbial data. This allowed us to assign benthic categories to these sites and extrapolate that the microbial communities found at these sites are somehow representative of the larger extent of the GBR. We have now added this explanation early in the methods section to establish the framework for the study. Please see lines 789-800 pasted as response to last comment. These classes are now used throughout the whole manuscript as a framework to summarize microbial communities.
11	Page 27: “after rarefying” - it would be nice to know to what read depth you rarefied in each region.	This was shown in Suppl. Table 1, but it is now explicitly mentioned in the text. Please see lines 932-933: “Alpha-diversity estimators (richness, Shannon and Chao diversity) were calculated after rarefying (to 25,000 reads)...”
12	Page 27: NMDS doesn’t “assess dissimilarity”. It is a visualization and dimensional reduction tool.	We now refer to “visualize community structure”. Please see lines 940-941: “Non-metric Multidimensional Scaling (NMDS – function “metaMDS” of vegan package ⁷⁵) limited to two dimensions was used to visualize the microbial community structure.”
13	Page 27: Why use both ANOSIM and PERMANOVA? Pick and justify one.	We now utilise PERMANOVA and justify this as a robust test for this data type.
14	Page 8: “which 16 were considered informative for microbial communities based on current literature” - Figure S1 lists 16, but doesn’t establish how they were chosen as being “informative for microbial communities”. Figure 2 does not hint at 16 variables. Make	These 16 variables are known potential drivers of microbial community variation. Although other variables may influence community structure, they are not frequently reported in biogeochemistry studies, precluding comparative analysis. We have revised this section to make this clear. Please see lines 203-207:

	this entire sentence much more crystal clear, as it also doesn't mesh with your modeling exercise with LDA.	“Modelled estimates of the environmental conditions of surface seawater retrieved from the eReefs hydrodynamic and biogeochemical model (GBR1, https://research.csiro.au/ereefs/models/model-outputs/gbr1/) for the microbial case study (n=37) and LTMP (n=109) sites (see Fig. 1), covered 16 environmental variables known as potential drivers of microbial community variation (see Fig. 2 and Supplementary Fig. 1).”; and a complete list of variables in the methods in lines 842-846: “Specifically, surface seawater temperature, salinity, total chlorophyll a, dissolved inorganic carbon (DIC), nitrogen (DIN) and phosphorus (DIP), ammonium (NH₄), nitrate (NO₃), dissolved organic carbon (DOC), nitrogen (DON) and phosphorus (DOP), total carbon (TC), nitrogen (TN) and phosphorus (TP), total suspended solids (TSS) and the vertical attenuation coefficient of light (K_{d_490}) were retrieved.”
15	Page 8: “Results on environmental variation across the GBR are detailed in the supplementary materials” means nothing. What are you rtying to say?	This sentence has been removed. These results are now presented in the main text. Please see new Fig. 2 and lines 208-218: “Reefs within distinct categories differed considerably in their prevalent surface seawater conditions (Fig. 2). Overall, organic and inorganic nutrients decreased in concentration with increasing distance from the shore (Fig. 2 and Supplementary Fig. 1; In-MA and In-Porites > Mid-Mixed and Out-Tab > Out-Soft and Out-Digit). The exceptions were the inorganic nitrogen variables (DIN, NH₄ and NO₃), which, together with chlorophyll a, peaked at midshelf reefs, particularly in the Out-Tab reef category. Superimposed on this inshore to outershell trend, there was strong seasonal variation. However, this effect tended to lose its influence towards outershell categories (Out-Soft and Out-Digit). Out-Soft and Out-Digit were devoid of seasonal effects, with the exception of temperature differences between the wet and dry seasons. Inshore reef categories in contrast, showed strong seasonal differences for all variables measured.”
16	Figure 2A does not explain the variables on the biplot arrows - they are just codes. These should be interpretable names of variables and explained in the legend and text.	We have restructured this section of the manuscript and the PCA plots have been removed.
17	Figure 2B can be moved to a supplement. It doesn't help inform	We agree and have now moved it to the supplementary (Suppl. Fig. 3a)

	the reader of anything but the accuracy of the LDA model.	
18	Figure 2c could be replaced by color coding the microbial sampling sites according to their modeled Mellin categories on the map in Figure 1. Then in Figure 1 make it abundantly clear that the microbial samples are modeled while the LTMP sites are measured. This will also be more informative than the proportions in Figure 2c.	We have opted to move this fig to the supplementary section together with fig 2B mentioned above as this is part of the LDA modelling exercise (please see new Suppl. Fig. 3b). Color-coding of the microbial sampling sites according to their modelled categories, as suggested by the reviewer, is already available in Fig. 6 of the main manuscript (previous Fig. 5). To address the point about modelled vs measured data we have modified the legend of Fig. 6 to explicitly state that reef category attributed to microbial samples is modelled in our study. Please see lines 540-541: “Note that panel a also includes the microbial sampling sites, now color-coded according to their modelled reef categories following methodology described in this study;”
19	Figure 2d is simply uninformative, plus it is showing a PCA of modeled information which is misleading.	We have removed this figure as suggested.
20	I recommend that Figure 2 be moved to three supplements (a, b, and c+d) and that you provide means for the various water chemistry parameters in the Mellin categories in both the measured (LTMP) and LDA-modeled (microbial) datasets. This will be much more digestible than the PCA plots.	We appreciate the reviewer’s excellent suggestion. Abiotic data was available and retrieved from the eReefs platform for both LTMP and microbial datasets (i.e., data modelled by eReefs). As suggested, we now provide a new figure with means (plus SD) for every parameter (please see new Fig. 2). We decided to provide a single figure with all sites combined (LTMP and microbial sites combined), but in the supplementary (Supplementary Fig 1a and b) we provide the same data split according to dataset (LTMP vs microbial). As trends are similar for both datasets, we decided to include a single figure in the main manuscript (averaging across all sites, LTMP and microbial combined). Notably, data was extracted from eReefs using the exact same approach and for the same time period. Please see new Fig. 2 and methodological description in lines 846-851: “For each of these 16 environmental variables, data spanning the period Jan 2015 - Jan 2018, were extracted for each 1x1 Km grid cell matching the n=37 sites that are part of the analysed case studies (further referred to as “microbial” sites) and n=109 reference sites included in the GBR-wide analysis of Mellin et al.³⁵ (further referred to as “LTMP” sites) using an adapted R script (https://github.com/sammatthews990/eReefs_Fr

		adeetal2019 .”
21	Page 12: This discussion is great, and is consistent with many other studies of microbes across coral reefs. For example, the work of Yeo et al 2013 Plos ONE, Nelson et al 2011 ISMEJ, McCliment et al 2011 ISMEJ, all track these gradients and discuss the common organisms found nearshore.	We have incorporated this additional literature within the discussion. Please see lines 595-597: “These bacterial community patterns suggest that terrestrial and riverine influences impose selective processes under which a narrower array of microbial taxa can thrive^{43,44}”; lines 610-612: “Changes in bacterioplankton community structure related to freshwater runoff have also been described for Hawaiian reefs⁴⁸”; and lines 773-775: “These interactions drive higher spatial heterogeneity of bacterial communities in inshore reefs as compared to more homogeneous outershelf and open ocean sites⁶⁷”.
22	Page 13: Dimension reduction is not shown in Figure S5. Are you saying that the PCAs in Figure S6 (and maybe Figure 2) only use 8 variables (or less in Figure 2) because you have done some kind of dimensional reduction (presumably with ggpairs?). This should be clear and easy to follow.	We thank the reviewer for pointing this out as dimensional reduction was done with ggpairs. Following a previous comment by the reviewer about the use of PCA (comment #8.3 on normality assumption), we have restructured this section and performed the necessary re-analyses. PCA was modelled using z-scores (standardized data) after ensuring that data was normally distributed (with transformations where needed). Please see lines 413-415: “After dimensional reduction based on pairwise correlations between variables (Supplementary Fig. 5), 8 non-collinear variables were visualized by PCA (Supplementary Fig. 6a; first two components explained 61.3% of variation in the dataset).”
23	Page 13: I don't follow the sentence “Replacing environmental variables by reef category and season explained 32.5% and 7.1% of variation, respectively.”	We have removed this sentence for clarity. This was just a side note and not part of the main analyses.
24	I find Page 14-15 to be a useful discussion and Figure 4 does not do it justice. I recommend that Figure 4A be amended with some simpler graphical depictions of some of the patterns discussed in this paragraph, as these are the crux of the paper and are lost in Fig 4. If the authors can use clustering or other methods to synthetically represent and distill some of the patterns great, otherwise just simple wet/dry	We thank the reviewer for this excellent suggestion. In the revised version we provide the recommended figure as new Fig. 5.

	correlations between environment and selected taxonomic groups would be insightful.	
25	Figure 5b: the pie charts should be sized (or annotated with a number) according to the summed relative abundance of indicator organisms at each site. This will help clarify that indicators of offshore sites (Pro) are very abundant in those sites, while indicators of nearshore sites tend to be rarer organisms.	We agree with the reviewer and have annotated the figure with numbers representing the summed relative abundance of indicator organisms at each site. Please see new Fig. 6.
26	Also, for the sake of the sanity of coral microbiology, please annotate the figure itself to clarify that these are relative abundances of indicator organisms only, and that this does not represent the bacterioplankton communities.	We have annotated the figure as requested. Please see new Fig. 6.
27	Figure 4b seems like it should be earlier in the paper, and it is not clear if it is from LTMP or the modeled microbial sites. See my earlier comments requesting something like this. What purpose does it serve here?	We agree with the reviewer and have made this new Fig. 2 in the revised manuscript.
28	Page 21: replace “nutrient enrichment” with “nutrient concentration”	Change made as requested. Please see lines 706-707: “The Prochlorococcaceae : Synechococcaceae relative abundance ratio represents a potential indicator for the contribution of nutrient concentration in coral reef waters.”
29	Page 21: replace “driving” with “associated with”	Change made as requested. Please see line 708-710: “(…) with a four-fold increase in nitrogen and phosphate concentrations associated with Synechococcus dominance from 9-15% to 64-66% of the cyanobacterial population ¹² .”
30	Page 21: “Additional indices could be developed to monitor eutrophication of GBR waters, for instance by including particular lineages (of Prochlorococcus) with different substrate affinities.” Because Pro is not strongly associated with nearshore habitats, I would argue that this would be better suited to exploring Syn and/or other microbial taxa for “nutrient-affiliated ecotypes” or whatever. Just because Pro has been studied	We have modified the sentence to illustrate that these indices were not meant specifically for Prochlorococcus , rather this was used as an example. Please see lines 726-728: “Additional indices could be developed to monitor eutrophication of GBR waters, for instance by including particular lineages (e.g., Prochlorococcus or Synechococcus) with different substrate affinities.”

	in this capacity does not make it a good indicator for terrestrial nutrient inputs.	
--	---	--

REVIEWERS' COMMENTS:

Reviewer #1 (Remarks to the Author):

The revised manuscript has responded accordingly to my concerns. The analytical method was clarified, the restriction of dataset was concerned, and the discussion was confined properly. I am also aware of the limitation and difficulty in doing more comprehensive analysis. I have no further questions.

Reviewer #2 (Remarks to the Author):

The authors have done a commendable revision and addressed all of my extensive recommendations thoroughly and intelligently. I consider the revised manuscript acceptable for publication in Communications Biology.